



# Part 1: Zonal gradients in phosphorus and nitrogen acquisition and stress revealed by metaproteomes of *Prochlorococcus* and *Synechococcus*

Claire Mahaffey[1*]; Noelle A. Held[2,3,4*]; Korinne Kunde[5,6]; Clare Davis[1†]; Neil Wyatt[6]; E. Matthew R. McIlvin[2],
Malcolm S. Woodward[7]; Lewis Wrightson[1]: Alessandro Tagliabue[1]; Maeve C. Lohan[6]; Mak Saito[2]

[1] Earth, Ocean and Ecological Sciences, University of Liverpool, UK L69 3BX
[2] Department of Marine Chemistry and Geochemistry, Woods Hole Oceanographic Institution, Woods Hole, USA

[3] Department of Environmental Systems Science, ETH Zürich, Zürich, Switzerland
[4] Department of Biological Sciences, Marine & Environmental Biology Section, University of Southern California, Los Angeles, CA, USA
[5] School of Oceanography, University of Washington, Seattle, USA
[6] Ocean and Earth Sciences, University of Southampton, UK SO14 3ZH
[7] Plymouth Marine Laboratory, UK PL1 3DH

[†] Current address: Springer Nature, London, UK

*Joint first authors and corresponding authors: mahaffey@liverpool.ac.uk, nheld@usc.edu

**Research Article**

**Short summary**

Marine primary production supports marine ecosystems and helps to regulate climate through carbon cycling.
The magnitude of productivity is underpinned by the availability of nutrient resources, such as nitrogen, phosphorus, iron, zinc and cobalt. Natural variation alongside anthropogenic activity has the potential to alter both the absolute and relative amount of nutrients available to marine microbes. To fully understand the impact of the evolving nutrient resource environment on marine primary productivity, we need to know how different marine microbes acquire nutrients, and which nutrients have the potential to limit productivity. In this study, we
used zonal gradients in nutrients, trace metals, biological activity and protein biomarkers representing phosphorus, nitrogen and trace metal acquisition and metabolism to better understand how two dominant picocyanobacteria, *Prochlorococcus* and *Synechococcus*, acquire nutrient resources in the surface subtropical ocean. Our suite of measurements agree on the occurrence of phosphorus stress for both *Prochlorococcus* and *Synechococcus* in the western Atlantic, but increases in proteins representing nitrogen, iron, zinc and cobalamin
metabolism in *Prochlorococcus* in the east where phosphorus biomarker proteins are lower indicates a switch in the nutrient resources controlling the growth of *Prochlorococcus* across the transect. Our study highlights the power of a combined discovery and targeted proteomics approach in providing species and even ecotype level information on nutrient acquisition and metabolism, which alongside measurements of states and rates, can be powerful tools in enhancing understanding of microbe metabolism in a changing climate.

**Abstract**

Ocean warming alongside changes to the natural and anthropogenic supply of key nutrient resources such as nitrogen, phosphorus and trace metals is predicted to alter the magnitude and stoichiometry of nutrients that are
essential for maintaining ocean productivity. To improve our ability to predict how marine microbes will respond to a changing nutrient environment, we need to better understand how natural assemblages of marine microbes acquire nutrients. We combined observations of natural zonal gradients across the North Atlantic subtropical gyre of the state of nutrient resources and microbial proteomes with biological activity rates, to investigate the factors influencing the distributions and nutrient acquisition strategies of the dominant picocyanobacteria,
*Prochlorococcus* and *Synechococcus*. Dissolved organic phosphorus decreased by more than a factor of two moving westward, while phosphate increased eastward with eastern boundary upwelling and dissolved iron increased westward with dust deposition. Picocyanobacterial populations diverged across the zonal transect with *Prochlorococcus* increasing in abundance westward, while maintaining numerical dominance throughout, and





while *Synechococcus* increased in abundance in the westward basin, implying a low phosphorus niche. We
analysed the zonal distribution of protein biomarkers representing phosphorus (PstS, PhoA, PhoX), nitrogen (P-II, UrtA, AmtB) and trace metal metabolism (related to iron, zinc and cobalt) alongside the response of phosphorus protein biomarkers to the addition of dissolved organic phosphorus with iron or zinc within incubation experiments. Rates of alkaline phosphatase alongside phosphorus protein biomarkers concur on more intense phosphorus stress in the western compared to the eastern subtropical Atlantic for both picocyanobacteria. Protein biomarkers for nitrogen, iron, zinc and cobalamin in *Prochlorococcus* increased to the east where phosphorus protein biomarkers were lower, indicating a transition to N stress and increasing role of trace metal resources in controlling *Prochlorococcus* growth. We use the diverging zonal patters in protein biomarkers, alongside the response of *Prochlorococcus* and *Synechococcus* to nutrient addition, to provide insight into the environmental controls on protein biomarkers of picocyanobacteria across the subtropical gyre. For example, the addition of DOP, Fe or Zn decreased PstS and PhoA in *Prochlorococcus* but increased PstS and PhoA in *Synechococcus*, implying divergence in regulation of phosphorus uptake or acquisition strategy. We postulate on the coinciding influences of upwelling, nitrogen fixation and atmospheric deposition on nutrient resources and controlling biogeography of picocyanobacteria. Together these biogeochemical and metaproteomic data imply a basin-scale transition from phosphorus stress in the west to nitrogen stress in the east within the picocyanobacteria on this zonal transect across the North Atlantic Ocean, with implications for productivity.

## 1. Introduction

Marine phytoplankton play a critical role in elemental cycling, supporting ecosystems and regulating climate. Global net primary productivity (NPP) is underpinned by availability of key nutrient resources, such as nitrogen (N), phosphorus (P), iron (Fe) and zinc (Zn) and many others. Over large ocean regions, such as subtropical gyres, concentrations of these key nutrients are chronically low in surface waters and often limit NPP. Enhanced stratification, induced by ocean warming, alongside changes to natural and anthropogenic supply of fixed N (Kim et al., 2014, Chien et al., 2016, Wrightson and Tagliabue 2021), P (Barkley et al., 2019) or Fe (Liu et al., 2022) to the global ocean are likely to perturb the magnitude and ratio at which nutrients are supplied to phytoplankton (Peñuelas et al., 2013), potentially expanding or intensifying nutrient limited ocean regions (Bopp et al., 2013, Chien et al., 2016, Lapointe et al., 2021). Detecting and understanding how nutrients regulate phytoplankton distribution, growth and activity is key to estimating the magnitude and direction of contemporary and future NPP, reducing uncertainty and assessing risks to ecosystem services (Tagliabue et al., 2021).

Evaluating the prevalence and impact of limitation by one or more nutrients on the growth and activity of phytoplankton in surface waters is typically done by detecting the response of the biological community to the experimental addition of nutrients, known as 'bioassays' (Mills et al., 2004, Moore et al., 2008, Mahaffey et al., 2014, Browning and Moore 2023) or comparing cellular and ocean resource inventories (Moore et al., 2013). Over the past decade, analysis of the genetic or protein makeup of marine phytoplankton to identify proteins indicative of nutrient acquisition and/or stress, termed biological markers or 'biomarkers' has accelerated due to the advances in 'omics' techniques (Saito et al., 2014, 2015, 2020, Rouco et al., 2018, Held et al., 2020, Ustick et al., 2021, Held et al., (submitted), Chappell et al., 2012). A recent evaluation of phytoplankton nutrient limitation at a global scale using a combination of approaches revealed agreement between incubation-based bioassays and genomic or proteomic analysis in identifying regions of N and/or Fe stress (Browning and Moore, 2023). However, identifying regions of P stress was more challenging, with disagreement between results from nutrient bioassays, implying a lack of P stress and genomic data, which identified large areas of P stress, especially for the



ecologically important picocyanobacteria, *Prochlorococcus*, in the low-phosphate ocean regions (Browning and Moore, 2023). This mismatch may be due to the flexibility in P acquisition strategies demonstrated by key marine phytoplankton (Scanlan et al 1993, Moore et al., 2005, Martiny et al., 2006, Martiny et al., 2009, Tetu et al.,

2009, Martinez et al., 2011). Under phosphate scarcity, phytoplankton can deploy an array of strategies to acquire alternative sources of P from dissolved organic phosphorus (DOP) including esters (Sebastian and Ammerman, 2009, Tetu et al., 2009), polyphosphate (Moore et al., 2005), phosphite (Martinez et al., 2011) and phosphonate (Ilikchyan et al., 2010). A hydrolytic metalloenzyme group, alkaline phosphatases, are responsible for cleaving P from esters (Hoppe, 2003). Enhanced activity of alkaline phosphatase (AP) has been used an indicator of P

limitation (Mahaffey et al., 2014, Su et al., 2023), although the substrate specificity (Srivastava et al., 2021), cellular localisation (Luo et al., 2009), AP allocation between ecotypes (Moore et al., 2005), uncertainty in the contribution of different phytoplankton groups to total enzyme activity (Held et al., (submitted; companion study to this manuscript))  and lack of knowledge on the efficiency of different AP enzymes raises uncertainties in its applicability. Collectively, the flexibility in P acquisition strategies, as well as the perceived ability of

*Prochlorococcus* to readily satisfy their P demands at ultra-low concentrations of phosphate (Lomas et al., 2012) has led to the idea that *Prochlorococcus* evade nutrient stress, particularly by remodelling their proteomes.

Comparing the physiological response of two ecologically important picocyanobacteria, *Prochlorococcus* and *Synechococcus,* to P stress demonstrates the complexity of deciphering resource limitation in mixed populations,

between species, or even between strains of the same species. *Synechococcus* possess genes encoding for a high affinity periplasmic phosphate binding protein (pstS) and transport system (pstABC), as well as genes encoding for proteins essential for accessing organic P via alkaline phosphatase (phoA) and phosphonatase (phnC, D, E, Scanlan et al., 1993, Moore et al., 2005, Tetu et al., 2009). When phosphate is scarce, *Synechococcus* has been shown to upregulate pstS, pstABC and phoA (Moore et al., 2005, Tetu et al., 2009), with a measurable increase

in AP activity (Moore et al., 2005), implying that expression of these genes is indicative of P starvation (Moore et al., 2005). However, clade specific variations in response to phosphate limitation have been observed *in situ* (Sohm et al., 2016) and in culture (Moore et al., 2005). While *Prochlorococcus* also possesses pstS and pstABC and has been shown to upregulate these genes alongside phoA under phosphate deplete conditions (Martiny et al., 2006), strain specific variations in its ability to access organic P also exist. For example, while the two most

prevalent high light (HL) clades, MED4 (HL1) and MIT9312 (HLII) can grow solely on phosphate, MED4 grows on a wider range of organic P compounds and dramatically increases AP activity when P starved compared to MIT9312 (Moore et al., 2005). In addition to species and clade specific responses across the microbial realm, AP enzymes are dependent on a metal co-factor, with Zn and/or cobalt (Co) required for the protein PhoA (Coleman, 1992) and Fe and calcium for the proteins PhoX and PhoD (Rodriguez et al., 2014, Yong et al., 2014). Although

the active sites of PhoA and PhoX in marine microbes have yet to be characterised, their metal requirements have been estimated assuming they are like the model organism, *Escherichia coli*. This dependency creates the potential for trace metal control on P acquisition via regulation of AP activity and thus Fe-P or Zn-P co-limitation (Browning et al., 2017, Mahaffey et al., 2014, ., et al submitted). Observations of an accelerating stoichiometry of Co in the western North Atlantic has led to hypotheses for the potential for Co use in oceanic alkaline

phosphatases too (Jakuba et al., 2008; Saito et al., 2017, Held et al., submitted). In culture studies,



*Prochlorococcus* and *Synechococcus* have been shown to have absolute requirements for Co but not Zn under replete P conditions (Saito et al., 2002; Sunda and Huntsman 1995; Hawco et al., 2020), but *Synechococcus* benefits from available Zn to produce AP under P scarcity (Cox and Saito 2013). Thus, knowledge of the phytoplankton community structure, alongside their nutritional preferences and enzyme characteristics is key in

deciphering nutrient limitation in the ocean.

The North Atlantic Gyre is heavily influenced by Saharan aeolian dust (Jickells, 1999), while the eastern basin borders the upwelling system off northwest Africa (Menna et al., 2016). Both upwelling and dust deliver scarce resources to the region, creating strong gradients in nutrients and trace metals (Gross et al., 2015; Sebastián et al., 2004; Reynolds et al., 2014, Kunde et al., 2019), influencing productivity (Moore et al., 2008)

and marine dinitrogen ($N_2$) fixation (Moore et al., 2009). Here, we exploit these strong natural gradients in nutrient and trace metal resources and biological activity to investigate nutrient acquisition strategies of natural assemblages of *Prochlorococcus* and *Synechococcus*. Alongside measurements of states, specifically nutrients, dissolved iron, zinc, cobalt and DOP and rates, including AP activity and $N_2$ fixation, we used a metaproteomic approach and quantified three proteins representing the high affinity phosphate binding protein, PstS, and two

alkaline phosphatases, PhoA and PhoX in *Prochlorococcus* and *Synechococcus* (Table 1). To support our investigation into P acquisition, we also considered three proteins indicative of N acquisition (P-II, UrtA, AmtB) and proteins involved in iron (ferredoxin), zinc (zinc peptidase and transporter) and $B_{12}$ (cobalamin synthetase) metabolism (Table 1). Firstly, we investigated the potential for *Prochlorococcus* and *Synechococcus* to be phosphorus-stressed in the subtropical Atlantic, challenging the view of avoidance of P limitation and

hypothesised zonal gradients in proteins would reflect nutrient stress. We also assessed the potential for N, Fe and Zn to control the zonal distribution of *Prochlorococcus* and *Synechococcus*. Secondly, we assessed the potential for P acquisition to be regulated by the availability of DOP, Fe and Zn or Co. We hypothesised that the distribution of PhoA and PhoX would be reflected in rates of AP and alongside Fe and Zn, the limiting trace metal. We augmented *in-situ* sampling with nutrient bioassays, complimentary to those reported by Held et al.,

(submitted), to further assess the potential for DOP substrate, alongside metals Fe and Zn to regulate AP activity and applied a quantitative proteomic approach targeting PstS, PhoA and PhoX only. Finally, we critically assessed our different approaches to delineate nutrient controls of the distribution and physiological strategies of *Prochlorococcus* and *Synechococcus*, highlighting challenges faced when bringing together biogeochemical measurements alongside 'omics (Saito et al 2024).


Table 1. Summary of the proteins targeted by metaproteome (all) and quantitative (*) protein analysis including their function and known characteristics.

| Protein name or family | Function and reported characteristics |
|---|---|
| PstS* | Periplasmic phosphate-binding protein. Induced under P-limiting conditions |
| PhoA* | Alkaline phosphatase: cleaves phosphorus from organic compounds. Zinc metalloenzyme Induced under P-limiting conditions |
| PhoX* | Alkaline phosphatase: cleaves phosphorus from organic compounds. Iron metalloenzyme. Regulation unknown |
| P-II | Nitrogen regulatory protein. Indirectly controls the transcription of glutamine synthetase gene glnA. |
| AmtB | Ammonium transporter channel. Transmembrane |



| UrtA | An ABC-type, high-affinity urea permease. Substrate binding protein |
|---|---|
| Ferredoxin | Iron metalloenzyme. Regulated by iron, more abundant under high iron conditions. |
| Zinc peptidase | Zinc metalloenzyme. Involved in proteolysis at the plasma membrane |
| Zinc transporter | Zinc metalloenzyme. ABC transporter, ATP-binding protein |
| Cobalamin synthetase | Cobalt metalloenzyme. Synthesis of cobalamin (vitamin B$_{12}$) |

## 2 Materials and methods

### 2.1 Sample collection from surface waters

Sampling was performed onboard the *RRS James Cook* (JC150) along a zonal transect between Guadeloupe and Tenerife at approx. ~22°N between 26[th] June and 12[th] August 2017 (Fig. 1a). Sea surface temperature (SST) was measured via the underway seawater system using Seabird sensors. Using a trace-metal clean towed FISH and a Teflon diaphragm pump (Almatec A-15), seawater samples were collected every 2 h, at a resolution of ~ 25 km, from ~ 3 m below the surface (Fig. 1a), with seawater flow terminating into a class-100 clean air-laboratory.

### 2.2 Biogeochemical states and rates

Using unfiltered seawater samples from the towed FISH, concentrations of nitrate plus nitrite (Brewer and Riley, 1965), phosphate (Kirkwood, 1989) and ammonium (Jones, 1991) were analysed onboard according to GO-SHIP nutrient protocols (Becker et al., 2020). Using filtered seawater from the towed FISH (Sartobran, Sartorius, 0.8/0.2 μm polyethersulfone membrane), concentrations of dissolved iron (Kunde et al., 2019) were measured onboard while concentrations of dissolved zinc (Nowicki et al., 1994) were determined at the University of Southampton. Concentrations of DOP were determined at the University of Liverpool using a modified version of Lomas et al. (2010), as described by Davis et al. (2019). Using unfiltered seawater from the towed FISH, rates of alkaline phosphatase were determined onboard every 4 h or ~ 50 km; Davis et al., 2019). *Prochlorococcus*, *Synechococcus* (or *Parasynechococcus*, Coutinho et al., 2016) and high and low nucleic acid bacteria (HNA and LNA, respectively) were enumerated every 2h at Plymouth Marine Laboratory using flow cytometry (Tarran et al., 2006). Surface ocean concentrations of chlorophyll *a* (on GF/F) were determined on every sample (Welschmeyer 1994). Concentrations of dissolved cobalt were measured in separate samples collected from 40m from 4 stations only using high resolution inductively coupled plasma mass spectrometry (HR-ICP-MS), preceded by UV-digestion and off-line preconcentration into a chelating resin (WAKO) at the University of Southampton (Rapp et al., 2017, Lough et al., 2019)

### 2.3 Global metaproteomic analysis

At 7 stations, McLane pumps were deployed to 15 m (see Table S1 for deployment details). Data from station 1 was omitted from this study due to significant riverine influence (Kunde et al., 2019). Pumps were fitted with a trace metal clean mini-MULVS filter head. Between 17 and 359 L of seawater was filtered through a 51 μm (Nitex), 3 μm (Versapor) and 0.2 μm (Supor) filter stack. Filters were immediately frozen at -80°C, with subsequent transportation and storage at -80°C. Protein biomarker analysis was conducted on the 0.2 μm filter,



representing the 0.2 to 3 µm particle fraction. Briefly, upon return to the laboratory, the total microbial protein was extracted using a detergent based method. The filter was unfolded and placed in an ethanol rinsed tube, then covered in 1 % SDS extraction buffer (1 % SDS, 0.1M Tris HCl pH 7.5, 10 mM EDTA), incubated at room

temperature for 10 mins, then at 95 °C for 10 mins, and then shaken at room temperature for 1 h. The extract was decanted and clarified by centrifugation before being concentrated by 5 kD membrane centrifugation to a small volume, washed in extraction buffer, and concentrated again. The total protein concentration was determined by BCA assay (kit) at this time. The proteins were precipitated in cold 50 % methanol 50 % acetone 0.5 mM HCl at 20 °C for one week, collected by centrifugation at 4 °C, and dried by vacuum. Purified protein pellets were

resuspended in 1% SDS extraction buffer and redissolved for 1 h at room temperature. Total protein was again quantified by BCA assay to assess recovery of the purification.

Extracted proteins were immobilized in a small volume polyacrylamide tube gel using a previously published method (Saito et al, 2014, Lux and Zhu 2005). LC-MS/MS grade reagents were used and all tubes were ethanol rinsed. The gels were fixed in 50 % ethanol, 10 % acetic acid, then cut into 1mm cubes and washed in

50:50 acetonitrile: 25 mM ammonium bicarbonate for 1 h at room temperature, then washed again in the same solution overnight. Next, the gels were dehydrated by acetonitrile treatment before protein reduction by 10 mM dithiothreitol treatment at 56 °C for 1 h with shaking. Gel pieces were rinsed in 50:50 acetonitrile: ammonium bicarbonate solution, then proteins were alkylated by treatment with 55 mM iodacetamide at room temperature for 1 h with shaking. Gels were again dehydrated by acetonitrile treatment and dried by vacuum. Finally, proteins

were digested by treatment with trypsin gold (Promega) prepared in 25mm ammonium bicarbonate at the ratio of 1:20 µg trypsin: ug total protein overnight at 37 °C with shaking. The next morning, any supernatant was decanted into a clean microfuge tube, and 50 µL protein extraction buffer (50 % acetonitrile, 5 % formic acid in water) was added to the gels, incubated for 20 mins, centrifuged and collected. The extraction was repeated and combined with the original supernatant. Peptides were concentrated to approximately 1 µg total protein per µL

solution by vacuum at room temperature. 10 µL or 10 µg were injected per analysis.

Global metaproteome analysis, which is conducted with no prior determined targets, was performed in Data-Dependent-Acquisition (DDA) mode using Reverse Phase Liquid Chromatography – active modulation – Reverse Phase Liquid Chromatography Mass Spectrometry (RPLC-am-RPLC-MS) (McIlvin and Saito 2021). RPLC-am-RPLC-MS involves two orthogonal chromatography steps, which are performed in-line on a Thermo

Dionex Ultimate 3000 LC system equipped with two pumps. The first separation was on a PLRP-S column (200 µm × 150 mm, 3 µm bead size, 300 Å pore size, NanoLCMS Solutions) using an 8 h pH 10 gradient (10 mM ammonium formate and 10 mM ammonium formate in 90% acetonitrile), with trapping and elution every 30 mins onto the second column. The second separation occurred in 30 min intervals on a C18 column (100 m × 150 mm, 3 µm particle size, 120 Å pore size, C18 Reprosil-God, Maisch, packed in a New Objective

PicoFrit column) using 0.1% formic acid and a 0.1% formic acid in 99.9% acetonitrile. The eluent was analyzed on a Thermo Orbitrap Fusion mass spectrometer with a Thermo Flex ion source. MS1 scans were monitored between m/z 380 and 1,580, with an m/z 1.6 MS2 isolation window (CID mode), 50 ms maximum injection time and 5 s dynamic exclusion time.

Resulting spectra were searched in Proteome Discoverer 2.2 with SequestHT using a custom DNA

sequence database consisting of over 30 genomes from cyanobacteria isolates and metagenomic data from the



Pacific and Atlantic oceans (including metagenomes from Metzyme and Geotraces cruise GA03). Annotations were derived using BLASTp against the NCBI non-redundant protein database. The corresponding protein FASTA file is available with the raw mass spectra files (see Supplement A). SequestHT parameters were set to +/1 10ppm for the parent ion, 0.6 Da for the fragment, with cysteine modification (+57.022) and variable

methionine (+16.0) and cysteine oxidation allowed. Protein identifications were made using Protein Prophet in Scaffold (Proteome Software) at the 95 % peptide confidence level, resulting in <1 % protein and peptide FDRs. Details of the peptides identified relative to protein name and organism can be found in Table S2 and the protein report and analytical details can be found in Supplement B.

**2.4 Quantitative proteomics analysis**

A small number of tryptic peptides were selected for absolute quantitative analysis in the samples from nutrient addition experiments (see section 2.5 for details) and were analysed as described by Held et al., (submitted). The amino acid sequence for the protein biomarkers quantified in this study (PstS, PhoA, PhoX) for *Prochlorococcus* and *Synechococcus* are summarised in Table S3 and peptide report and analytical details are found in Supplement

C.

**2.5 Nutrient bioassay experiments**

Trace-metal clean sampling and incubation protocols used to setup onboard bioassays are described in detail in the Supplement D.  Aliquots of Fe, Zn and Co solutions were added to unfiltered seawater to investigate metal

limitation of alkaline phosphatase and results are reported in Held et al., (submitted). Alongside these experiments, we added DOP alone or with Fe and Zn to investigate the potential for organic P availability to influence AP activity at stations 2 and 3 only, where concentrations of DOP were low (< 80 nM, Fig. 1a and e, Table S4), and the results are reported here. Trace-metal clean 20L carboys were triple rinsed with unfiltered seawater collected from 40m (to avoid contamination from the ship) via the FISH and filled and amended

accordingly (Table S4). We measured the change in phytoplankton biomass (chlorophyll *a*, abundance of *Prochlorococcus*, *Synechococcus*) and AP activity at the start of the incubations and after 48 h. We also quantified the concentration of proteins (PstS, PhoA and PhoX) as described in section 2.4 (Table S3). Incubations were conducted in triplicate. However, due to the biomass (therefore volume) required for protein analysis, we were unable to collect samples from three incubation bottles for further analyses. Instead, all

measurements were collected from two incubation bottles, except aliquots for determination of AP, which was collected from three incubation bottles. To compare the change in states or rates in treatments relative to the control, we considered a significant change in a property to occur when the mean of the property in the amended incubation was 2-times higher (or lower) than the mean control incubation. Incubations were conducted in a temperature controlled container set to a temperature measured at 40m (between 25 and 27°C) and with 12:12h

light:dark cycle simulated by LED light panels (Part no: LED-PANEL-300-1200-DW and LED-PANEL-200-6-DW, Daylight White, supplier Power Pax UK Limited).

**3. Results and Discussion**





**3.1. Zonal trends in states and rates:** SST decreased from ~ 28 °C in the west to ~ 25 °C in the east (Fig. 1b).

In the upper 10 m, phosphate increased from ~ 5 nM to 20 nM from west to east (Fig. 1c), whereas nitrate plus

nitrate (N+N, herein nitrate) ranged from < 10 nM to ~ 40 nM with no clear zonal trend (Fig. 1d). Ammonium

concentrations ranged from 3 to 21 nM, with the highest concentrations observed between stations 5 and 6 (Fig.

1f). From west to east, DOP increased 3-fold (from ~ 50 nM to ~ 150 nM, Fig. 1e) alongside a 4-fold decrease in

chlorophyll corrected AP activity (from > 2000 nmol P μg chl $a$ d$^{-1}$ to < 500 nmol P μg chl $a$ d$^{-1}$, Fig. 1g).

Concurrent zonal gradients of phosphate, DOP and AP activity supports previous findings (Mahaffey et al., 2014)

of an increase in AP activity as phosphate decreases (Fig. S1a), driving a decline in DOP (Fig. S1b).

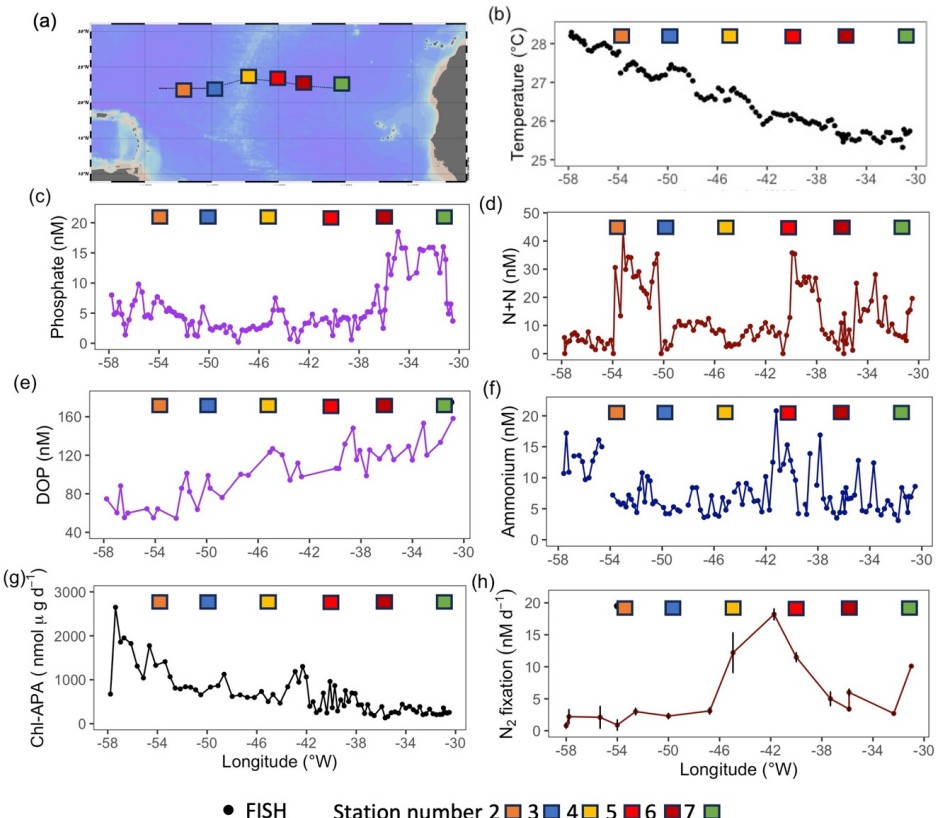

Figure 1. (a) Locations sampled during JC150 from the trace metal clean towed FISH (black circles) and stations
(coloured squares) and surface ocean properties including (b) sea surface temperature (ºC), (c) phosphate (nM),
(d) nitrate+nitrite (N+N, nM), (e) dissolved organic phosphorus (DOP, nM), (f) ammonium (nM), (g) chlorophyll
a – corrected rates of alkaline phosphatase (nmol P mg chl d$^{-1}$), (h) mean rates of dinitrogen (N$_2$) fixation (nM N
d$^{-1}$) with error bars as standard deviation of triplicate incubations. Note that data from JC150 Station 1 (test
station) has not been included in this manuscript due to the strong riverine influence (Kunde et al., 2019). Map
produced using Ocean Data View (ODV).

Dissolved Fe concentrations decreased from ~ 1.4 nM in the west to ~ 0.4 nM in the east (Fig. 2a), with higher Fe

in the west due to enhanced dust deposition from the Saharan dust source that is transported to the western

Atlantic Ocean (Kunde et al., 2019). Zn was variable throughout the transect, ranging from 0.04 to 0.8 nM (Fig.

2b). Cobalt was measured at 40 m and at 4 stations only and ranged from 11 pM at stations 2 and 3 to 13-13.9



pM at stations 4 and 7 (data not shown). Chlorophyll *a* concentration increased from ~ 0.05 µg L$^{-1}$ to ~ 0.15 µg L$^{-1}$ from west to east (Fig. 3a). *Prochlorococcus* cell abundance increased from ~ 5 x 10$^4$ cells mL$^{-1}$ in the west to 2.5 x 10$^5$ cells mL$^{-1}$ in the east (Fig. 3b), whereas *Synechococcus* cell abundance decreased from ~ 8 x 10$^3$ cells mL$^{-1}$ in the west to 1 x 10$^3$ cells mL$^{-1}$ (Fig. 3c). Both HNA and LNA bacterial abundance increased ~ 2-fold from

west to east (Fig. 3d and e, respectively). Rates of N$_2$ fixation were highest between stations 4 and 5 (12 to 18 nM d$^{-1}$) and were elevated in the east (3 to 10 nM d$^{-1}$) compared to the west (< 3 nM d$^{-1}$) (Fig. 1g). In addition, there was an increase in the abundance of key diazotrophs *Trichodesmium* and UCYN-A in the east relative to the west (Cerdan-Garcia et al., 2021).

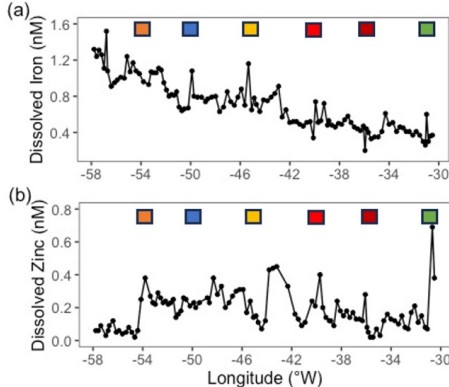


Figure 2. Zonal gradients in (a) dissolved iron concentrations (nM, from Kunde et al., 2019) and (b) dissolved zinc concentrations (nM). Samples captured from the towed FISH at ~ 7m. Coloured square represent stations sampled during JC150 (see Fig. 1 for station names).




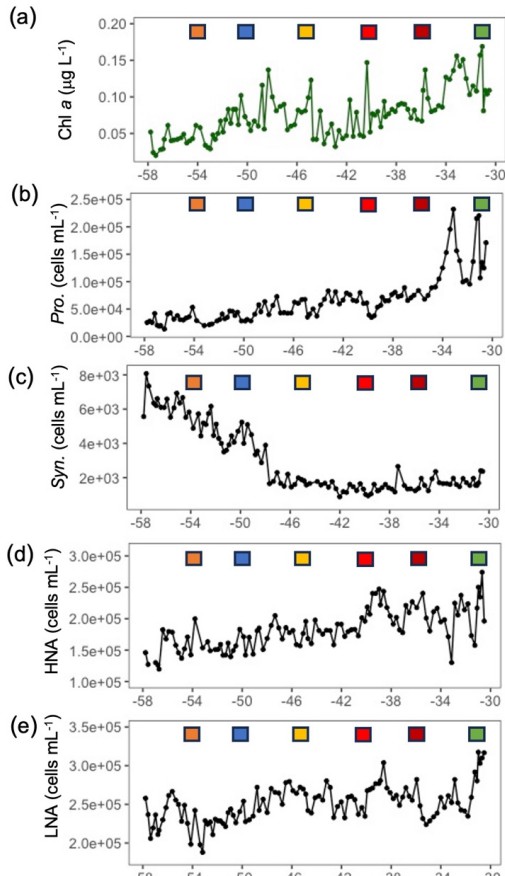

Figure 3. Zonal gradients in (a) chlorophyll a concentrations (µg chl L$^{-1}$) and the abundance of (b) *Prochlorococcus* (cells mL$^{-1}$), (c) *Synechococcus* (cells mL$^{-1}$), (d) high nucleic acid bacteria (HNA, cells mL$^{-1}$)
and (e) low nucleic acid bacteria (LNA, cells mL$^{-1}$). Samples captured from the towed FISH at ~ 7m. Coloured square represent stations sampled during JC150 (see Fig. 1 for station names).

The zonal gradients in the hydrography, nutrients, biological rates and cyanobacteria create two

contrasting regions in the west (west of 46°W or west of station 4) and east (east of 46°W or east of station 4),

allowing a quantitative comparison of key characteristics between station 2 at 54°W and station 7 at 31°E (Fig.

1a, Table 2). In the west, concentrations of dissolved Fe and ammonium, averaged over mixed layer, were 3 to 4-

fold higher, APA was 4-fold higher and *Synechococcus* abundance was 2-fold higher compared to the east. In

contrast in the east, phosphate, DOP and chlorophyll *a* were 2 to 4-fold higher, *Prochlorococcus* abundance was

6-fold higher, rates of N$_2$ fixation were 3-fold higher (excluding maximum rates at stations 4 and 5) and

*Trichodesmium* and UCYN-A abundances were 2 and 71-fold higher (Table 2). Against a background of low

phosphate concentrations along the entire transect, we would expect PstS to be prevalent throughout the basin,

indicating phosphate stress. At the same time, we would expect a strong west-east gradient in protein biomarkers

indicative of higher AP in the west to match the observed trends in rates of APA. We would expect prevalence of

PhoX in this Fe-rich basin and Fe-stress biomarkers to be more prevalent in the east where dissolved Fe

concentrations were lower.



| Properties higher in the west (-fold) | Properties higher in the east (-fold) |
|---|---|
| Iron (3) | Phosphate (4) |
| Ammonium (4) | DOP (3) |
| APA (4) | Chlorophyll (2) |
| $V_{max}/K_m$ (5) | *Prochlorococcus* (6) |
| *Synechococcus* (2) | $N_2$ fixation rates (3) |
| | *Trichodesmium* (2) |
| | UCYN-A (71) |
| | |
| *Prochlorococcus* -Phosphate binding protein, PstS (2) | *Prochlorococcus* - Nitrogen regulatory protein, PII (1.3) |
| *Prochlorococcus* -alkaline phosphatase, PhoA (7) | *Prochlorococcus* - Ammonium transporter, AmtB (1.7) |
| *Synechococcus* -alkaline phosphatase, PhoA (29) | *Prochlorococcus* -Urea permease, UrtA (1.6) |
| SAR11-alkaline phosphatase, PhoA (24) | *Prochlorococcus* -Ferredoxin (9) |
| Total *Synechococcus* protein (1.3) | *Prochlorococcus* -Zinc peptidase (1.3) |
| | *Prochlorococcus* -Zinc transporter (4) |
| | *Prochlorococcus* - Cobalamin synthetase (5) |
| | SAR11- alkaline phosphatase, PhoX (4) |
| | Total *Prochlorococcus* protein (1.6) |

Table 2. Summary of states, rates and protein biomarkers that are higher in the west (left hand column) or east (right hand column) of the transect. The numbers in brackets represent the approximate -fold difference between west and east west. Properties not reported (e.g. dissolved zinc, *Syn*-UrtA) displayed no clear difference between west and east.


### 3.2 Phosphorus acquisition by *Prochlorococcus*

*Prochlorococcus* (HLII) specific P proteins PstS and PhoA (*Pro*-PstS and *Pro*-PhoA, respectively) were almost 2-fold and 7-fold higher in the west relative to the east (Fig. 4a, Table 2), whereas there was no clear zonal trend in PhoX (*Pro*-PhoX, Fig. 4a). For *Pro*-PstS and *Pro*-PhoA, the zonal trends were consistent across other clades of

*Prochlorococcus* for PstS (Fig. S2a), PhoA (Fig. S2b) and PhoX (Fig. S2c). Note that total *Prochlorococcus* protein (reported as total spectral counts, Fig. 4d) agreed with the zonal trend *Prochlorococcus* cell abundance (Fig. 3b) suggesting that untargeted metaproteomics analysis can capture trends in microbial community structure. Thus, higher *Pro*-PstS and *Pro*-PhoA in the west where there was lower *Prochlorococcus* reflects a physiological response to the nutrient environment rather than reflecting changes in biomass.




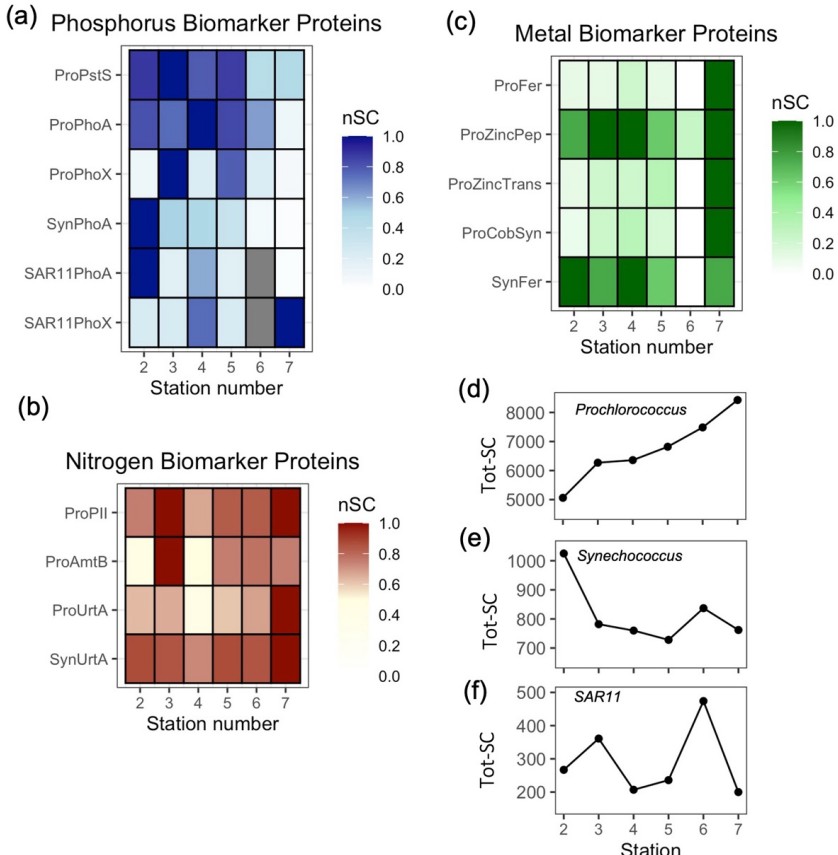

Figure 4. Zonal gradients in the spectral counts (SC) of biomarker proteins in *Prochlorococcus* (Pro-), *Synechococcus* (Syn-) and SAR11 for (a) Phosphorus biomarker proteins; PstS, PhoA and PhoX, (b) Iron, zinc and cobalt biomarker proteins; Ferredoxin (Fd) and Zinc peptidase (ZincPep), Zinc transporters (ZincTrans) and Cobalamin Synthetase (CobW) (c) Nitrogen biomarker proteins: PII, AmtB and UrtA and (d) total protein for *Prochlorococcus*, (e) *Synechococcus* and (f) SAR11, presenting an independent measure of biomass. See Table 1 for details of the protein functions. nSC represents normalized spectral counts, which represents the spectral counts normalized to the maximum value of each protein across 6 stations. Tot-SC represents the sum of all normalized spectral counts for *Prochlorococcus*, *Synechococcus* or SAR11

*Prochlorococcus* cell abundance (and in turn, total *Prochlorococcus* protein) was negatively correlated with *Pro*-PstS (p=0.03, Fig. 5a), *Pro*-PhoA (p=0.035, Fig. 5b) and APA (p=0.001, Fig. 5d) and positively correlated with DOP (p=0.007, Fig. 5c). *Pro*-PstS was negatively correlated with DOP (p=0.03, Fig. 5e) and both *Pro*-PstS and *Pro*-PhoA were positively correlated with AP activity (p=0.013 and p=0.057, respectively, Fig. 5f and g). In agreement with biogeochemical signatures for P stress, proteins, *Pro*-PstS and *Pro*-PhoA were more prevalent in the west (Fig. 4a, Table 2) and significantly positively correlated to AP activity (Fig. 5f and g). However, *Prochlorococcus* abundance was lower in the west compared to the east (Fig. 3b), and negatively related to overall AP activity, *Pro*-PstS and *Pro*-PhoA and positively related to DOP concentrations. Together these data imply increased P stress of *Prochlorococcus* in the west compared to the east. This was again evident in greatly increased P-stress biomarkers in the west, despite decreasing *Prochlorococcus* abundance, demonstrating the change in biomarkers is not simply due to changing biomass.



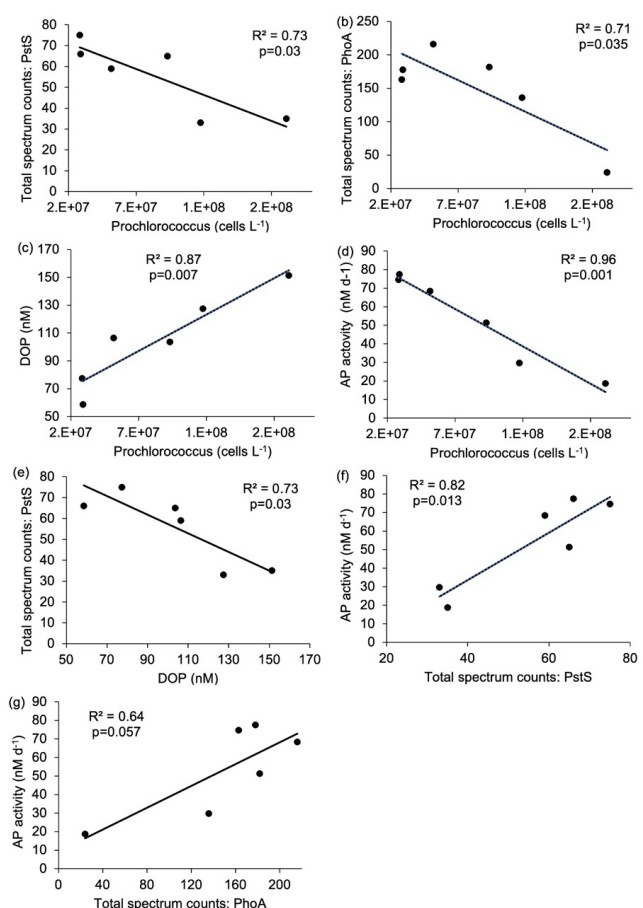

Figure 5. Relationship between (a) *Prochlorococcus* cell abundance (cells $L^{-1}$) and Pro-PstS (total spectrum counts), (b) *Prochlorococcus* cell abundance (cells $L^{-1}$) and Pro-PhoA (total spectrum counts), (c)

Prochlorococcus cell abundance (cells $L^{-1}$) and dissolved organic phosphorus (DOP, nM), (d) *Prochlorococcus* cell abundance (cells $L^{-1}$) and rates of alkaline phosphatase (APA, nM $d^{-1}$), (e) DOP and Pro-PstS, (f) Pro-PstS and APA and (g) Pro-PhoA and APA. Results are linear regression as reported as $R_2$ value and p-value. Relationships shown in (a) to (f) are considered statistically significant as $p < 0.05$.

Within the nutrient bioassays, there was evidence that DOP limited phytoplankton growth and activity at station

2, expressed as an increase in mean chlorophyll *a* (from 0.075 to 0.120 µg $L^{-1}$) and mean rates of APA (3.03 to

9.70 nM $d^{-1}$), especially after the addition of DOP+Fe (* denotes a 2-fold or greater increase relative to the

control, Fig. 6a, see Table S5 for raw data). No significant changes to growth or activity were observed at station

3 (Fig. 6b, Table S5). In the bioassays, there was a decline in the concentration of *Pro*-PstS and *Pro*-PhoA after

the addition of DOP (Fig. 6a and b), implying protein production is repressed in the presence of elevated DOP.

This experimental observation is supported by in-situ observations as *Pro*-PstS and *Pro*-PhoA both decreased to

the east (Fig. 4a) where DOP and phosphate were elevated in surface waters (Fig. 1c and e). This DOP effect was

likely the result of DOP conversion to phosphate and negative regulation of the Pho operon that controls both

PstS and PhoA rather than DOP directly interacting with the regulatory system. *Pro*-PhoX increased more than 2-





fold after the addition of DOP alone and DOP+Zn at station 2 (Fig. 6a) but the lack of knowledge of controls on

PhoX mean it is premature to interpret this observation. One caveat is that the abundance of *Prochlorococcus*,

gleaned from flow cytometry, declined in all experiments and also eastward along the zonal transect (Fig. 6a and

b, Table S5), meaning it is unclear if the decline in protein biomarkers, *Pro*-PstS and *Pro*-PhoA, in experiments

and surface waters, was a physiological response to elevated DOP, or due to a decline in *Prochlorococcus*

biomass. However, knowledge of the dominant *Prochlorococcus* clades in the Atlantic Ocean (Johnson et al.,

2006) alongside selection of protein markers to target specific clades means that our ability to interpret at the

ecotype level resolution both in experiments, as well as over large spatial transects is rapidly improving (Saito et

al., 2015), highlighting the benefits of 'omics applications to study microbial biogeography.

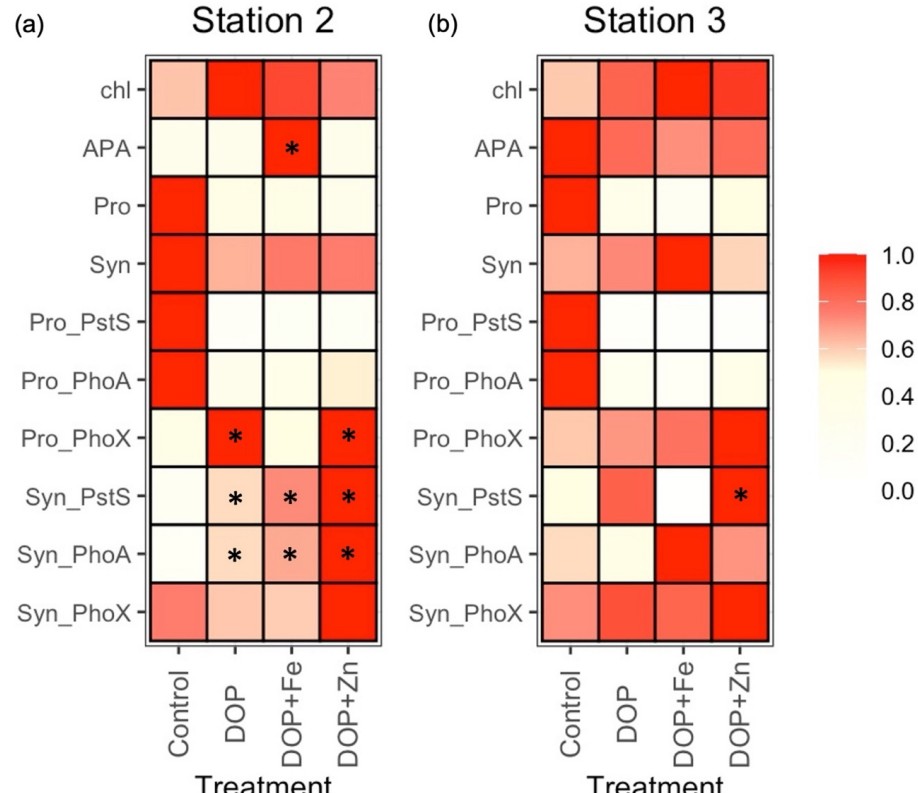


Figure 6. Fractional (scale 0 to 1) change in states, rates and individual proteins in *Prochlorococcus (Pro_)* and
*Synechococcus* (*Syn*-) after the addition of dissolved organic phosphorus (DOP), DOP and iron (DOP+Fe) and
DOP and zinc (DOP+Zn) at Station 2 (a) and Station 3 (b) for chlorophyll *a* (chl), rates of alkaline phosphatase
activity (APA), *Prochlorococcus* (Pro), *Synechococcus* (Syn) and protein biomarkers PstS, PhoA and PhoX.
Coloured squares represent the mean of duplicate or triplicate samples and are normalised as the fraction of the
maximum of that property in each experiment. See Table S4 for a description of the experiments and Table S5
for raw data for all properties. * denotes a 2-fold or more change in the mean property relative to the control.

For example, culture studies reveal that *Prochlorococcus* HLII, the dominant clade in the oligotrophic subtropical

ocean, can use ATP but not other organic P sources and shows minimal increase in APA once P starved, possibly



due to the lack of the regulatory genes (*psiP1*, *ptrA*) responsible for activating genes that respond to P limitation (Moore et al., 2005). This contrasts with HL1 (MED4), which can grow on a variety of organic P sources, increases AP activity (up to 8-fold) above their measurable constitutive activity when grown on organic P as its

sole source of P relative to phosphate (Moore et al., 2005), suggesting upregulation of genes encoding for AP in the presence of external organic P (Moore et al., 2005). Using the global metaproteome, we detected an increase in *Prochlorococcus* ecotypes HL1 (Fig. S3a) and HLII (Fig. S3b) from west to east in agreement with the increased cell abundance and total *Prochlorococcus* protein. The contribution of HLI to total ecotype counts increased slightly from around 6 to 8% (Fig. S3c). Previous studies have found HLI (MED4) to be more

prevalent in the eastern Atlantic (Zinser et al., 2007) but the subtle increased observed in this study may reflect the depth sampled (5 to 15m) as HLI abundance typically increases with depth (Johnson et al., 2006, Zinser et al., 2007).

### 3.3 Phosphorus acquisition by *Synechococcus*

PhoA in *Synechococcus* (referred to as *Syn*-PhoA) was 29-fold higher in the west compared to the east (Fig. 4a, Table 2). *Syn*-PhoA was negatively correlated with DOP (p=0.006, Fig. 7a) and positively correlated with APA (p=0.018, Fig. 7b), but unlike *Prochlorococcus*, there was no correlation between cell abundance and proteins, DOP or AP (Fig. S4). Note that the concentration of other *Synechococcus* P-related proteins, *Syn*-PstS and *Syn*-PhoX, were not detected in the metaproteome during the time of sample collection and using the methods applied

but may have been present albeit at low concentrations. The higher *Synechococcus* abundance in the west compared to the east coincided with higher APA (Fig. 1f), higher *Syn*-PhoA (Fig. 4a) and higher total *Synechococcus* protein count (reported as total spectral counts, Fig. 4e).

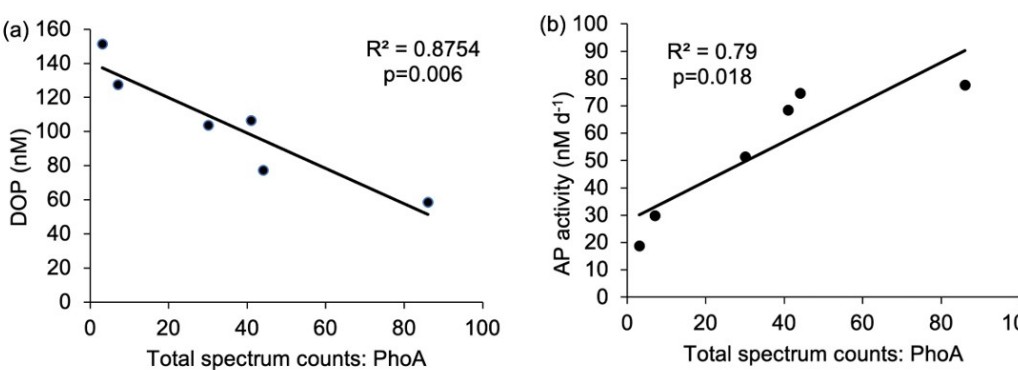

Figure 7. Relationship between (a) *Synechococcus* PhoA (total spectral counts) and concentrations of
DOP (nM) and (b) *Synechococcus* PhoA (total spectral counts) and alkaline phosphatase activity (AP, nM d$^{-1}$). The R$^2$ value and p-values are reported. p<0.05 indicates that the relationship is statistically significant.

The addition of DOP alone, or with iron (DOP+Fe) or zinc (DOP+Zn) during 48-h bioassays caused
*Synechococcus* abundance to decline by 23 to 35% at station 2 (Fig. 6a, Table S5). However, at station 2, the mean concentration of *Syn*-PstS increased by 2.7-, 3.5- and 4.7-fold after the addition of DOP, DOP+Fe and DOP+Zn, respectively (Fig. 6a, Table S5) and the mean concentration of *Syn*-PhoA increased by 3.6-, 4.3- and




6.4-fold after the addition of DOP, DOP+Fe and DOP+Zn, respectively (Fig. 6a, Table S5). It is unclear why
production of both *Syn*-PstS and *Syn*-PhoA was stimulated after the addition of DOP+Fe at station 2, and above

that of DOP alone considering PhoA having Zn or Co, and not Fe as metal co-factors. However, replication was
low (n=2) and variability between replicates was high, so it was not possible to unpick this result statistically. At
station 3, *Synechococcus* abundance increased by 12% and 53% after the addition of DOP and DOP+Fe
respectively but decreased after the addition of DOP+Zn (Fig. 6b, Table S5). The change in protein concentration
after nutrient additions was less pronounced at station 3 (Table S5). *Syn*-PstS increased by 1.8- to 2.1-fold after

the addition of DOP and DOP+Zn, respectively (Fig. 6b, Table S5), whereas *Syn*-PstS decreased by 90% after the
addition of DOP+Fe (Fig. 6b, Table S5). *Syn*-PhoA increased by 1.2- to 1.7-fold after the addition of DOP+Zn
and DOP+Fe, respectively, but decreased by 30% after the addition of DOP only. There was no notable change in
*Syn*-PhoX after the addition of DOP only, or with DOP+Zn and DOP+Fe, with *Syn*-PhoX increasing or
decreasing by 20 to 40% at both stations (Fig. 6b, Table S5).


These independent results from both in-situ measurements and bioassays converge to imply that Syne*chococcus*
is highly dependent upon organic P accessed via APA and can produce *Syn*-PstS and *Syn*-PhoA in the presence of
DOP and Zn to increase P acquisition when phosphate is low. The capacity of *Synechococcus* to increase AP
activity in presence of organic P (Waterbury et al., 1986) and produce proteins PstS and AP under low phosphate

conditions in the presence of Zn (Cox and Saito 2013), as observed in culture studies, supports our zonal trends in
proteins alongside results from bioassays.  Higher AP activity and prevalence of *Syn*-PhoA occurred in a region
of chronically low DOP and phosphate concentrations, implying that *Synechococcus* was P stressed. Using
enzyme kinetic bioassays, we detected higher AP enzyme efficiency in the west (Fig. 8a), which was positively
correlated with *Syn*-PhoA (p=0.017, Fig. 8b). We therefore speculate that *Syn*-PhoA can efficiently cleave P from

natural DOP compounds even at the low DOP concentrations observed in the west.

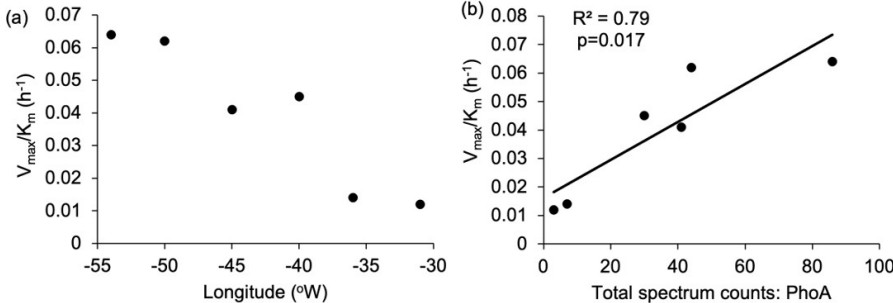

Figure 8. Enzyme efficiency for alkaline phosphatase was calculated as the ratio between $V_{max}$ and $K_m$ ($h^{-1}$); (a)
zonal gradient in enzyme efficiency, indicating higher enzyme efficiency in the western compared to the eastern

subtropical Atlantic and (b) positive significant (p=0.017) relationship between *Syn*-PhoA and enzyme efficiency.

We observed a similar west-east gradient in PhoA for SAR11 (Fig. 4a), an abundant aerobic chemoheterotrophic
alphaproteobacterial that contributes to LNA bacterial counts (Fig. 3e). While the abundance of both HNA and
LNA increase from west to east (Fig. 3d and e, respectively), also indicated by the increase in total SAR11



protein (Fig. 4f), SAR11-PhoA decreased 24-fold from west to east, whereas SAR11-PhoX increased 4-fold from
west to east (Fig. 4a, Table 2). The reason for the switch from PhoA in the west to PhoX in the east for SAR11 is
unclear, especially as Fe was higher in the west relative to the east. As above, we speculate that PhoA is efficient
at cleaving P from DOP under low P conditions, and thus SAR11 may strategically use PhoA under low P
conditions prevalent in the west of our transect. Culture studies reveal that organic P is an important source of P

to SAR11, meeting up to 70% of its cellular P requirement even when phosphate is non-limiting (Grant et al.,
2019) and thus the zonal patterns in SAR11-PhoA and PhoX likely reflect the preferential acquisition of DOP
over phosphate.  This further supports the premise that PhoA is an indicator of DOP acquisition across marine
microbial taxa.

Results from bioassays would suggest that *Syn*-PhoA should increase to the east as DOP concentration increase.
Instead, we observed an eastward decline in *Synechococcus* abundance and *Syn*-PhoA alongside the increase in
DOP, which suggests that other factor(s) affected the P acquisition strategy and growth of *Synechococcus*.
Averaged over the upper 40m, the mean Zn concentration declined from ~ 0.35 nM in the west to 0.15 nM in the
east and may have limited *Syn*-PhoA activity. The dependence of PhoA on Zn in *Synechococcus* has been

observed in culture with the production of PstS and PhoA only occurring at low phosphate as well as in the
presence of Zn (Cox and Saito 2013). Held et al., (submitted) observed more than a 2- to 4-fold fold increase in
*Syn*-PstS and *Syn*-PhoA after the addition of Zn or Co relative to the control, with the addition of Fe also
stimulating an increase in *Syn*-PstS. This finding represents new field evidence for Co influencing AP (Held et
al., (submitted), implying that Co may effectively substitute Zn at the active site of PhoA within marine

cyanobacteria, consistent with trends observed in accelerating Co stoichiometry and APase abundances in the
North Atlantic Ocean (Saito et al., 2017). Competition for P, trace metals and other resources with other
plankton, including *Prochlorococcus* and $N_2$ fixers, may have stunted growth of *Synechococcus* in the east
relative to the western subtropical Atlantic.

**3.4. Influence of trace metals on alkaline phosphatase and associated protein biomarkers**
Although strong zonal trends in dissolved Fe were observed (Fig. 2a), the distribution of *Pro*-PhoX gleaned from
the metaproteome analysis (Fig. 4a) did not reflect iron availability. Instead, protein concentrations quantified in
the control of the nutrient bioassays reveal that for *Prochlorococcus*, PhoA concentrations were 2.7 to 4.7-fold
higher than PhoX (reported as the number of enzymes per litre seawater basis, Table 3) at stations, 2, 3 and 5,

where Fe was elevated, but *Pro*-PhoX was greater than *Pro*-PhoA at station 7 where Fe was lowest (Fig. 2a).

PhoX was not detected for *Synechococcus* in metaproteome analysis but was detected quantitatively at the start of
bioassay experiments (Table 3). Between stations 2 and 4, *Syn*-PhoX decreased from $21 \pm 6$ fmol $L^{-1}$ to $9 \pm 5$ fmol
$L^{-1}$ in line with a decline in dissolved Fe but increased again to $16 \pm 2$ fmol $L^{-1}$ at station 7 where Fe was lowest

(Fig. 2a).  However, the ratio between PhoA and PhoX was variable for *Synechococcus* (Table 3) and the pattern
in Syn-PhoA differed between the quantitative and metaproteome analysis, likely due to the different depth
horizons sampled (40m for experiments, 15 meters for metaproteome analysis).





Table 3: Concentration of proteins (fmol L$^{-1}$) PhoX and PhoA for *Prochlorococcus* and *Synechococcus* at the start of the nutrient bioassay experiments at stations 2, 3, 4 and 7 (see Fig. 1a for locations) to illustrate how the relative concentration and ratio of PhoA to PhoX differ between *Prochlorococcus* and *Synechococcus* and across the zonal transect.

| Protein biomarker | *Prochlorococcus* | | | *Synechococcus* | | |
|---|---|---|---|---|---|---|
| Protein conc. (fmol L$^{-1}$) | PhoX | PhoA | PhoA/PhoX | PhoX | PhoA | PhoA/PhoX |
| Station 2 control | $17 \pm 2$ | $45 \pm 14$ | 2.7 | $21 \pm 6$ | $7 \pm 1$ | 0.3 |
| Station 3 control | $16 \pm 11$ | $48 \pm 10$ | 3.0 | $15 \pm 2$ | $22 \pm 12$ | 1.8 |
| Station 4 control | $7 \pm 0.1$ | $31 \pm 13$ | 4.7 | $9 \pm 5$ | $8 \pm 3$ | 0.8 |
| Station 7 control | $8 \pm 2$ | $3 \pm 1$ | 0.4 | $16 \pm 2$ | $28 \pm 13$ | 2.4 |

These findings for two dominant picocyanobacteria contrast with biogeography of PhoA and PhoX in *Trichodesmium* (Rouco et al., 2018), with PhoX being enriched in the subtropical North Atlantic and PhoA being enriched in the subtropical North Pacific oceans, consistent with the associated trace metal availability. To our knowledge the systems biology of the PhoX enzyme is poorly understood compared to that of PstS and PhoA, where the latter is known to be regulated by phosphate and zinc (Martiny et al., 2006; Tetu et al., 2009; Cox and

Saito, 2013), but it is unclear what regulates PhoX in cyanobacteria. There was no consistent or significant increased phytoplankton growth, increase in *Prochlorococcus* and *Synechococcus*, or AP activity after the addition of Fe, Zn or Co alone (Held et al., submitted). However, at station 2 in the west, *Syn*-PhoA increased 6 and 7-fold after the addition of Zn ($38 \pm 0.56$ fmol L$^{-1}$) and Co ($47 \pm 6.8$ fmol L$^{-1}$) relative to the control ($6.7 \pm 1.5$ fmol L$^{-1}$), respectively. At station 4 over the mid-Atlantic ridge, *Pro*-PhoX increased 8-fold upon addition of

Co relative to the control. At station 7 in the east, *Pro*-PhoX also increased over 2-fold upon Fe addition (to $18 \pm 2.6$ fmol L$^{-1}$) relative to the control ($8.2 \pm 2.4$ fmol L$^{-1}$, Held et al., submitted). Results from these bioassays illustrate the potential for a direct metal control on AP, lending support to the hypothesis for local, albeit patchy metal-phosphorus co-limitation in the subtropical North Atlantic (Jakuba et al., 2008, Mahaffey et al., 2014, Saito et al., 2017, Browning et al., 2017).

Alongside metal control of AP, there were gradients in proteins relating to iron, zinc and B$_{12}$ metabolism in *Prochlorococcus*. Ferredoxin increased from west (1 to 2 spectral counts) to the east (9 spectral counts) alongside zinc transporters (1 to 3 spectral counts in the west, 9 in the east) although there was no clear zonal trend in zinc peptidases (Fig. 4c, Table 2). We consider these zinc protein annotations in *Prochlorococcus* as putative because this transporter annotated in the *Prochlorococcus* genome has not been validated for Zn

transport, and alignment-based transporter annotations tend to have difficulty in discerning cognate metal use. Moreover, under phosphate-replete conditions, *Prochlorococcus* does not have an obligate Zn requirement (Saito et al., 2002), and Zn is highly toxic to a Pacific Ocean strain of *Prochlorococcus* (Hawco et al., 2018), contributing to the uncertainty of the role of zinc in *Prochlorococcus*. The protein annotated as CobW, is a member of the COG0523 family implicated in metal chaperone functions (Edmonds et al 2021), and Co

chaperone for B$_{12}$ synthesis (Young et al., 2021). While CobW is an abundant protein among the ~20 genes involved in cobalamin biosynthesis, there are currently no known biomarkers for cobalt or zinc metabolism in



*Prochlorococcus*, with studies producing negative results (Hawco et al., 2020). However, in our data, CobW increased from the west (2 to 7 spectral counts) to the east (22 spectral counts at station 7). While the eastward increase in three independent proteins coincides with an increase in *Prochlorococcus* cell abundance and total protein spectral counts, the 4 to 9-fold increase in ferredoxin, zinc transporter and cobalamin synthetase is greater than the increase in total protein for *Prochlorococcus* (1.6-fold) implying a regulated molecular increase in response to resource limitation or competition, rather than reflecting a change in biomass only. For *Synechococcus*, there were no clear trends in ferredoxin across the transect (Fig. 4c). Note that flavodoxin was infrequently detected across this surface transect. We surmise that for these picocyanobacteria, the lack of understanding of the regulatory system for PhoX, means it is difficult to predict the trace metal controls of the multiple APs in-situ, and in particular what drives the picocyanobacteria to switch between producing PhoA and PhoX. It may be that organic complex forms of Co or Zn regulate PhoX, and the appropriate laboratory and field experiment remains to be conducted. Finally, deconvoluting the P acquisition strategies alongside the biogeography of picocyanobacteria and their ecotypes *in situ* is further complicated by other environmental and physiological factors that can influence their growth. Additional mechanistic laboratory studies are needed to contribute to interpretation of picocyanobacterial alkaline phosphatases.

**3.5. Controls on biogeography of *Prochlorococcus* and *Synechococcus* in the subtropical North Atlantic**

With its small cell size, physiological plasticity, low P demand and potential resilience to P stress, *Prochlorococcus* is considered an excellent competitor in oligotrophic environments. How this translates into numerical abundance (stocks of cell concentrations) and ecological competitiveness (relative fitness) can be examined on the basin-scale zonal gradient in this study. We hypothesise on two ecological scenarios for this region. First, based on its oligotrophic niche, we predict that *Prochlorococcus* abundance should be higher in the oligotrophic low P west compared to the east, where phosphate and DOP were elevated. Second, upwelling in the eastern Atlantic delivers nitrate, phosphate and other nutrients to surface waters, enhancing biomass, productivity and production of dissolved organic nutrients (Mahaffey et al 2004). When upwelled waters are transported laterally into the oligotrophic gyre, we predict that *Prochlorococcus* maintains numerical dominance over other organisms. Organic nutrients are also transported laterally but due to the bioavailability of different organic pools, lateral supply of organic P is more effective than the lateral supply of organic N (Mahaffey et al., 2004), thus potentially relieving P- but enhancing N stress in our study region. Here, we use our untargeted metaproteomics data to explore factors beyond P acquisition that may control the biogeography of *Prochlorococcus* and *Synechococcus,* specifically the role of upwelling, N sources, dust and its constituents.

Our results are consistent with the second scenario above, where *Prochlorococcus* was numerically more abundant in the east following the trend of autotrophic community as indicated by chlorophyll a, and maintained numerical dominance over *Synechococcus* throughout the transect (Fig. 3b). Interestingly, while *Synechococcus* thrives in relatively nutrient-enriched environments, as observed in the east, the abundance of *Synechococcus* was actually higher in the west relative to the east (Fig. 3c).

The notion that phosphorus uptake by *Prochlorococcus* is so efficient as to prevent it from experiencing P stress does not appear to be supported by observations on this transect. DOP concentrations decreased 2-fold



from east to west and concurrent with organic P assimilation and depletion was a decline in *Prochlorococcus* abundance, and increased deployment of multiple P-stress acquisition systems. It appears the switch to organic P as the primary P source bears metabolic and ecological costs. The westward increase in *Synechococcus* (while still much lower in absolute numbers than *Prochlorococcus*) implies it may have a low P (inorganic and

organic) niche relative to picoeukaryotes, consistent with numerous studies that have observed high AP production by *Synechococcus* (Torcello-Requena et al., 2024).

The basin scale population changes in picocyanobacterial can also be considered from the perspective of N nutrition. *Prochlorococcus* are major players in the microbial loop and thus the availability of recycled nutrients such as ammonium and urea, driven by basin-scale processes, may influence their biogeography.

Although marine *Synechococcus* and some *Prochlorococcus* strains have the genetic makeup to assimilate nitrate (Martiny et al., 2009, Berube et al., 2015, Dominguez-Martin et al., 2022), nitrate accounts for < 5% of their total N demand, and instead ammonium and urea are the dominant N sources (Casey et al., 2007, Painter et al., 2008, Berthelot et al., 2018). Surface ocean nitrate (< 40 nM, Fig. 1d) and ammonium (< 20 nM, Fig. 1f) concentrations were low, with a 4-fold decrease in ammonium from west to east, and a maximum ammonium concentration

coinciding with the highest rates of $N_2$ fixation (Fig. 1h). Corresponding with these chemical observations, *Prochlorococcus* protein biomarkers P-II, ammonium transporter AmtB and urea transporter UrtA increased eastward (30-70%, Fig. 4b and Table 2). We postulate that the eastward increase in these proteins, especially urea transporter UrtA (also consistent across different clades, see Fig. S2d) was indicative of increasing N stress towards the eastern Atlantic in contrast to increasing P stress in the western Atlantic.

The North Atlantic is a region of enhanced $N_2$ fixation owing to the supply of iron-rich dust (Moore et al 2009) and *Prochlorococcus* (and *Synechococcus*) are prime beneficiaries of N exuded from $N_2$ fixers (Caffin et al., 2018). Enhanced rates of $N_2$ fixation and higher ammonium concentrations (Figures 1h and 1f, respectively) were observed in the middle of the zonal transect (38 to 46°W, between stations 4 and 5). *Prochlorococcus*-AmtB and UrtA were lowest at these stations (Figure 4b), likely reflecting alleviation of N stress as

*Prochlorococcus* benefited from N exudates, as observed in the North Pacific gyre (Saito et al 2013, Saito et al 2015). However, despite $N_2$ fixation rates being ~3-fold higher in the east compared to the west (Table 2), the overall increase in *Prochlorococcus* N stress biomarkers in the east indicates that this species may not have been the main beneficiary of this process.

For *Synechococcus*, the urea transporter UrtA spectral counts were more than 5 times higher than for

*Prochlorococcus* and were constant across the transect, implying *Synechococcus* was N stressed throughout the transect (Fig. 4b), likely due to its larger cell size and less efficient surface-area to volume ratio for nutrient acquisition (Chisholm 1992). We did not detect P-II and AmtB (or NtcA) in the metaproteome of *Synechococcus*, perhaps because *Synechococcus* was 5 to 10 times less abundant in the metaproteomes compared to *Prochlorococcus*.

In addition to N and P, where both *Prochlorococcus* and *Synechococcus* had the potential to benefit from freshly fixed N and the changes in P speciation and availability along the zonal transect, we can consider the trace element controls that could have impacted their biogeography. Upwelling in the eastern Atlantic may have delivered Fe and other trace metals to surface waters, with lateral transport potentially driving zonal gradients. However, dissolved Fe was low in the east and trace metal proteins increased relative to total proteins towards the



east, implying Fe, Zn and Co stress for *Prochlorococcus*, likely driven by resource competition with the autotrophic community, which was also high in the east, as indicated by chlorophyll *a*. However, in summer 2017, it is more likely that the strong zonal gradients in surface dissolved Fe concentration were driven by high deposition of aerosol dust (Kunde et al., 2019). Large amounts of coarse-grained dust are deposited by both dry and wet deposition in the eastern Atlantic, with lower amounts of finer-grained dust deposited by wet deposition

in the west during summer (van der Does et al., 2021). Wet deposition is efficient at delivering Fe to the ocean (Schlosser et al., 2014) but also other constituents including trace metals (e.g. copper and molybdenum) and macronutrients (e.g. nitrate, Baker et al., 2007, Powell et al., 2015, Benaltabet et al., 2023). Alongside the lack of nutrients, especially P, elevated dust deposition may have reduced the *Prochlorococcus* abundance in the west relative to the east due to greater copper toxicity, biological agents or other constituents from aerosol dust (Mann

et al., 2002, Herut et al., 2005, Hill et al., 2010, Rahav et al., 2020). *Synechococcus* is copper-resistant (Mann et al., 2002) and thus higher *Synechococcus* in the west may have been stimulated by nitrogenous nutrients deposited by dust aerosols (Paytan et al., 2009, Mackey et al., 2012).

Lastly, trophic interactions and nutrient recycling were also likely playing a role in determining patterns in the picocyanobacterial; *Synechococcus* in the east coincided with higher bacterial biomass (LNA and HNA,

Figures 3d and e), perhaps because *Prochlorococcus* stimulates heterotrophs, so they are successfully competing with *Synechococcus* for scarce nutrients (Calfee et al., 2022).

**3.6. Protein biomarkers as indicators of nutrient status in picocyanobacteria**

Increased prevalence of protein biomarkers PstS and PhoA in the west compared to the east, alongside the

positive relationship between *Pro*-PstS, *Pro*-PhoA and *Syn*-PhoA with AP activity and negative relationship with DOP support previous findings that these protein biomarkers are indicators of P stress in *Prochlorococcus* (Moore et al., 2005, Martiny et al., 2006, Reistetter et al., 2013) and *Synechococcus* (Scanlan et al., 1993, Tetu et al 2009) and challenges the view that growth of picocyanobacteria are insensitive to nutrient availability. However, the change in the concentration of these biomarkers after the addition of DOP in nutrient bioassays

differed between *Prochlorococcus* and *Synechococcus* (Fig. 6, Table S5). For *Prochlorococcus*, the addition of DOP reduced the concentration of *Pro*-PstS and *Pro*-PhoA, and increased PhoX (Fig. 6, Table S5) after the 48-h incubation period relative to the control. In contrast, for *Synechococcus*, the addition of DOP increased *Syn*-PstS and *Syn*-PhoA, with no change in PhoX (Fig. 6, Table S5). When the per cell protein content was calculated (with the caveat that the protein is clade specific yet likely targeted a major ecotype, whereas cell abundance represents

all cells), the same pattern is observed indicating that the change in protein biomarkers was not driven solely by the change in cell abundance but was rather being regulated in response to environmental conditions (Fig. S5). For *Prochlorococcus*, the PstS and PhoA per cell decreased and PhoA per cell increased 48-h after the addition of DOP, relative to the control (Fig. S5a). For *Synechococcus*, the PstS and PhoA per cell increased 48-h after the addition of DOP relative to the control, with an increase in PhoX (Fig. S5b). The divergence in response of the

same protein biomarkers to the same substrate implies that the regulatory pathway for these proteins differs between *Prochlorococcus* and *Synechococcus,* and/or that strain specific differences in quantified proteins is complicating our interpretation response of proteins across different strains. Below, we discuss both possibilities.



The cellular regulation of P responsive proteins in the marine cyanobacterium can provide insight into the

dynamics of abundance patterns and related metal requirements for AP. There is a Pho regulon within *Prochlorococcus* that controls genes associated with P acquisition in low-P environments such as pstS (phosphate transporter) and phoA and includes the genes for the two-component regulatory system itself (phoB and phoR, Martiny et al., 2006). The phoX gene in *Prochlorococcus* is found within a genomic island with other P stress responsive genes implying regulation by the pho regulon (Kathuria and Martiny, 2011). In contrast, a two-tiered

phosphate response system has been characterized in marine *Synechococcus* using model strain WH8102, where the PhoBR regulator controls pstS using a Pho box, consistent with experimental observations of responses to phosphate  (Tetu et al., 2009; Cox and Saito, 2013), and another regulator PtrA appears to operate above it, controlling one of the phoA phosphatase copies, Zn transport, and various other cellular processes (Ostrowski et al., 2010). The gene neighbourhood containing phoA (SYNW2391) in *Synechococcus* is also located near efflux

transporter and close to the ferric uptake regulator, Fur. To our knowledge, regulation of PhoX and its interaction with PhoA regulation in the marine picocyanobacterial is not well understood, but analysis of the gene neighbourhood in the model organism *Prochlorococcus* sp. NATL1A reveals that phoX is not within the phoA neighbourhood and is in the vicinity of a putative manganese transporter. For *Synechococcus* (WH8102), the position of phoX (SynW0120) is like *Prochlorococcus*, that is not within the phoA neighbourhood but close to

rod proteins, sugar uptake systems and hydrolytic enzymes responsible for homocysteine and adensine production (adenosylhomocysteinase). The separation of phoA and phoX within the genome in both *Prochlorococcus* and *Synechococcus* implies their regulation is distinct in the different organisms. While the different metal uses of these two APs implies that they may be regulated by metal as well as phosphorus availability, the specific regulatory system that may allow this in the marine picocyanobacterial are to our

knowledge still unknown. Results from the bioassays demonstrate this regulatory complexity, as demonstrated by the opposing response of PhoA and PhoX to the addition of DOP and metals during bioassays and between picocyanobacteria. Together our results are consistent with hypotheses and prior observations highlight a role for Fe and/or Zn switch to control APs within natural populations of cyanobacteria (Mahaffey et al., 2014, Browning et al., 2017, Rouco et al., 2018).


During the nutrient bioassays, proteins were quantified by detecting peptides with an amino acid sequence that was specific to strains and clades of *Prochlorococcus* and *Synechococcus* (Table S3, Supplement C). For *Prochlorococcus*, the peptide sequences targeted up to 5 strains, but focused on the HLII clade, meaning that comparisons between protein were comparable within the HLII clade and particularly within strain MIT9314,

providing reassurance in the interpretation of the response in nutrient bioassays. For *Synechococcus*, the peptide sequence for PhoA and PhoX targeted WH8102, representing clade III but the peptide sequence for PstS targeted RCC307, representing clade X (Table S3), meaning that the response of these three protein biomarkers is being compared across clades. While there is a positive correlation in the geographic distribution of clades III and X, typically co-occurring in warm oligotrophic waters, and a negative correlation with phosphate (Sohm et al.,

2016), RCC307 does not possess the same putative alkaline phosphatase genes as WH8102 (likely PhoA, see Tetu et al., 2009). This mismatch in targeted strains and clades means that interpretation of the response of



*Synechococcus* (and perhaps *Prochlorococcus*) to the addition of DOP needs to be treated with some caution as the physiology and regulatory pathways of protein production are better understood.

**4.0. Conclusions**

In the vast subtropical ocean gyres, the abundant picoplankton species, *Prochlorococcus* and *Synechococcus*, are responsible for > 60% of primary productivity and thus are key for the functioning of ecosystems and biogeochemical cycling. Both species are considered excellent competitors for nutrients, even at low concentrations, due to their small cell size, flexible nutrient demands, and ability to deploy a variety of nutrient

acquisition strategies (Lomas et al, 2021, Moore et al., 2005, Scanlan et al., 2009). Using these traits alongside rising temperature and enhanced surface ocean DOP recycling (White et al., 2012), results from statistical niche models predict an increase in the biomass of *Prochlorococcus* and *Synechococcus* over the coming decades largely driven by rising temperatures (Flombaum et al., 2013, Flombaum and Martiny, 2021). This contrasts with Earth system models, which project a decline in biomass and net primary production by up to 20% by 2100 in the

subtropical gyres under high emissions scenarios due to reduced nutrient supply and shifts to smaller cells (Bopp et al., 2013, Dutkiewicz et al., 2013, Chust et al., 2014, Tagliabue et al., 2021). Unlike in models, small celled functional groups encompass both *Prochlorococcus* and *Synechococcus*, who we have shown to deploy a range of strategies to acquire nutrients across a diverse resource landscape that encompasses inorganic and organic nutrient pools. How these findings integrate up to shape the response of primary productivity in response to

climate change is a major gap in knowledge, which requires a new generation of ecological-biogeochemical models parameterised based on the latest findings. Addition of N due to anthropogenic activity may intensify and expand P-stress in the future. Additionally, micronutrients, such as Fe and Zn are essential for key biological processes such as photosynthesis, $N_2$ fixation and perhaps also AP, and their supply to the ocean is predicted to change due to anthropogenic activity (Liu et al., 2022). If P-stress ultimately does increase, phytoplankton will be

under selective pressure to be more competitive for P, either by acquiring phosphate at chronically low concentrations or using metal-requiring alkaline phosphatases to access organic P. This underscores the need to understand the environmental and distinct physiological factors controlling the growth of these key marine picocyanobacteria to drive more mechanistic realism in future global models.

This study exploited natural gradients in nutrient resources created by upwelling in the east and dust deposition in the west. Combining states, rates and 'omics approach, akin to the aspiration of the developing 'BioGeoSCAPES program (Saito et al., 2024), we studied the nutrient acquisition strategies for *Prochlorococcus* and *Synechococcus* in-situ and using nutrient bioassays, with a focus on P. Using protein biomarkers alongside biogeochemical signatures for nutrient stress, we concluded that *Prochlorococcus* and *Synechococcus* were P-

stressed in the western Atlantic and *Prochlorococcus* was N-stressed in the eastern Atlantic, with *Synechococcus* showing signs of N-stress throughout the transect. Our findings are generally consistent with prior metagenomic observations on basin scale contrasts in N and P stress for *Prochlorococcus* in the Atlantic Ocean (at medium level, Ustick et al., 2021). There was evidence for trace metal control on alkaline phosphatase but the response of protein biomarkers to the addition of organic P, Zn and Fe differed between *Prochlorococcus* and *Synechococcus*

(also see Held et al., submitted), highlighting that the functions and systems biology of alkaline phosphatase



regulation differs across the organisms and for different environmental stimuli. This indicates that laboratory characterization of protein biomarkers will be useful for defining the regulation and function not only at the species level, but also across strains within species.

Under future climate scenarios, stratification, aerosol dynamics, $N_2$ fixation and the bioavailability of
organic P are predicted to change (e.g. White et al., 2012, Chien et al., 2016, Wrightson and Tagliabue, 2020, Buchanan et al., 2021), all with the potential to perturb the availability of already scarce nutrient resources in the oligotrophic gyres. To identify and quantify the future trajectory of *Prochlorococcus* and *Synechococcus* under future ocean scenarios, a holistic view that considers the species and strain specific strategies used to access resources, alongside representation of large scale forcings are required. We have shown here that there is utility
in combining biochemical assays with untargeted and targeted omics approaches to reveal these patterns, generate hypotheses that can be tested in controlled laboratory experiments, and improve predictions of marine microbiology and biogeochemistry in a changing ocean.

**Competing interests**: The authors declare no competing interests

**Data availability:** All new data are provided in the Supplement or are available from the British Oceanographic
Data Centre (BODC) with the following DOIs: Size-fractionated iron measurements (https://doi.org/10.5285/8a1800cc-b6a6-30ea-e053-6c86abc0c934), inorganic nutrients, alkaline phosphatase, DOP, chlorophyll, flow cytometry: https://doi.org/10.5285/284a411e-2639-93de-e063-7086abc0e9d8), Experiment D (https://doi.org/10.5285/1e9c4caa-b936-fc7c-e063-7086abc06ff6). The mass spectrometry proteomics data have been deposited to the ProteomeXchange Consortium via the PRIDE partner repository with
the dataset identifier PXD054252 and 10.6019/PXD054252

**Supplement.** Supplementary information is provided as individual files and 1 zip file. There are 5 supplements including Supplement A (Fasta file), Supplement B (protein file), Supplement C (peptide file) and Supplement D (trace metal clean protocols for nutrient bioassays). In the zip file, there are 5 supplementary tables provided as
spreadsheets (Table S1 to S5) and 5 supplementary figures (Fig. S1 to S5).

**Author contributions:** CM, MCL and AT acquired the funding from NERC. CM and MCL led the research cruise. CD conducted AP measurements. KK conducted Fe measurements. NW conducted zinc measurements. MSC conducted nutrient measurements. LW conducted $N_2$ fixation measurements. CM, MCL, CD, KK, LW and
NW conducted the large volume incubation experiments. KK and NL conducted the quantitative proteomics analysis and MM analysed samples using mass spectrometry at WHOI. NL conducted the global metaproteome analysis. CM and NH wrote the manuscript with significant contributions from MS, ML, CD, KK and AT.

**Acknowledgements:** The authors would like to thank the officers and crew of the *RRS James Cook* for the
successful research cruise, JC150. This research was supported by the Natural Environment Research Council (NE/N001079/1, awarded to CM and AT, NE/N001125/1 awarded to ML), Simons Foundation Grants 1038971 and BioSCOPE, Chemical Currencies of a Microbial Planet (CCOMP) NSF-STC 2019589 to M.A.S, an ETH Zurich Career Seed Grant to N.A.H, and the USCDornsife College of Arts and Sciences. K.K. was supported by
Graduate School of the National Oceanography Centre Southampton (UK) the Simons Foundation (award 723552) during the writing process. The authors would also like to thank Alastair Lough and Clément Demasy for the dissolved cobalt measurements.



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
