# Peer review of "Proteomic and biogeochemical perspectives on cyanobacteria nutrient acquisition: Part 1: Zonal gradients in phosphorus and nitrogen acquisition and stress revealed by metaproteomes of *Prochlorococcus* and *Synechococcus"

_EGUsphere, 2024_

## Author Response (AR1)

**Comment from Zhou Liang (26th March 2025)**

The linkage between marine phosphorus and trace metal cycling in the modern ocean has received attention in recent years, resulting in an increased recognition of the importance of metal-dependent DOP acquisition by diverse marine microbes, as recently summarized by Duhamel et al. (2022). However, these interactions are complex and not fully understood. This work provides valuable insights into these interactions by integrating geochemical observations, metaproteome analysis and bioassay experiments. They found a decrease in trace metal stress and an increase in phosphate stress toward the west in the North Atlantic. This trend coincides with DOP consumption to the west. The observed response of DOP consumption to the availability of trace metals and phosphate is consistent with previous prediction based on geochemical evidence (Liang et al., 2022). It also shows the potential of metaproteomics as a powerful tool for unrevealing microbial processes in the modern ocean.
Reference:

Duhamel, S., Diaz, J. M., Adams, J. C., Djaoudi, K., Steck, V., & Waggoner, E. M. (2021). Phosphorus as an integral component of global marine biogeochemistry. *Nature Geoscience*, *14*(6), 359-368.
Liang, Z., Letscher, R. T., & Knapp, A. N. (2022). Dissolved organic phosphorus concentrations in the surface ocean controlled by both phosphate and iron stress. *Nature Geoscience*, *15*(8), 651-657.

We thank the Dr. Zhou Liang for their positive comments on the manuscript and appreciate the recommendations for literature on this topic. We have now included these in the manuscript.

**Comment from referee no. 1 (19th May 2025)**
Review of Mahaffey et al., 2025 Zonal gradients in phosphorus and nitrogen acquisition and stress revealed by metaproteomes of *Prochlorococcus* and *Synechococcus*

This manuscript utilises a protein biomarker approach to assess nutrient stress in marine picocyanobacterial populations across a transect in the North Atlantic Ocean. I have long been a fan of utilising protein biomarkers as a means of telling us directly what cells are experiencing in their in situ environment but I also know that you need to interpret this data very carefully. Hence whilst I acknowledge the tremendous amount of work that has gone into this manuscript I also believe the authors need to be very clear with both the benefits and limitations of these studies – and with some mention of this in the abstract warranted at least (since the authors allude to differences in regulation of specific genes between Synechococcus and Prochlorococcus, but this can also be extended to differences in regulation within a genus (i.e. between Pro ecotypes or Syn clades) and potentially between strains (as well as importantly whether there is indeed differential presence/absence of genes in ecotypes/clades which can complicate interpretation of subsequent metaproteomics data). Please see my specific comments on this manuscript below:

We have updated the abstract with a note on the caveats associated with ecosytems and clades.

Line 39: microbial metabolism
Microbe changed to microbial

Line 53 please delete 'and' at the end of the line
Done

Line 62: patterns
Done

Line 100: Ostrowski et al., 2010 ISME J should also be included here

Done

Line 110: The Lomas et al., 2012 reference is either missing or incorrect
Correct reference was Lomas et al 2014. Now updated

Line 115: please delete for i.e. encoding a …
Done

Line 117: please delete for
Done

Line 117 and Line 119: a new high affinity phosphatase (psip) was recently described and should be cited in this section (i.e. line 119) as well as Ostrowski et al 2010 since this is regulated by PtrA

Thank you for pointing this out. We have now referred to Torcello-Requena et al., 2024 and Ostrowski et al 2010 in this section.

Line 128: I think the authors should be up-front here about the fact that PhoA for Prochlorococcus and Synechococcus is not only not biochemically characterised but also atypical compared to the E. coli PhoA. The implication that PhoA requires zinc like the E. coli enzyme might be wrong. Indeed, the atypical alkaline phosphatase present in freshwater Synechococcus (PCC7942), which has sequence similarities to SYNW2390 and SYNW2391 in Syn WH8102, is inhibited by zinc! (see Ray et al., J. Bacteriology 1991)

We agree with the reviewer that reliance on homology to model organisms and a lack of biochemical characterization of alkaline phosphatases in *Prochlorococcus* and *Synechococcus* can make it challenging to be 100% confident in the metal cofactor for the enzyme based on sequence alone. However, the most recent data supports that marine Syn PhoA is a Zn containing protein. In Cox & Saito 2013, PhoA increases in response to phosphate scarcity in *Synechococcus* sp WH 8102 but does not do so when no Zn is provided. In the same organism it was determined that znuABC Zinc transport system (*synw2479–synw2481)* is not regulated by the master-zinc regulator Zur but is regulated by phosphate (Ostorowski et al, 2010; Mikhaylina et al., 2022), again suggesting that zinc is used for phosphorous acquisition. Likewise, our own data supports the idea that PhoA is a Zn containing protein.

*In addition to species and clade specific responses across the microbial realm, AP enzymes are dependent on a metal co-factor, with Zn and/or cobalt (Co) required for the protein PhoA (Coleman,*
*1992) and Fe and calcium for the proteins PhoX and PhoD (Rodriguez et al., 2014, Yong et al., 2014). Although*
*130 the active sites of PhoA and PhoX in marine microbes have yet to be biochemically characterised, their metal requirements have been estimated assuming they are like the model organism, Escherichia coli and based on supporting evidence that the enzymes respond to the metals that they are expected to contain (Cox & Saito, 2013, Ostorwosky et al., 2010, Mikhaylina et al., 2022). However, homology-based annotation of enzymes is challenging and therefore the annotations herein should be considered putative.*

Line 133: what is this submitted paper?
Held, N. A., Kunde, K., Davis, C. E., Wyatt, N. J., Mann, E. L., Woodward, E. M. S., McIlvin, M., Tagliabue, A., Twining, B. S., Mahaffey, C., Saito, M. A., and Lohan, M. C.: Part 2: Quantitative contributions of cyanobacterial alkaline phosphatases to biogeochemical rates in the subtropical North Atlantic, EGUsphere [preprint], https://doi.org/10.5194/egusphere-2024-3996, 2025.

Line 136-7: Again, this is a suggestive sentence since this implies PhoA from picocyanobacteria requires zinc (which we don't know) – see comment above Edit in line with above statements

We have added the caveat regarding the trace metal (or zinc) dependence in response to the query at line 128 and believe this caveat covers this statement too.

Line 220: 25 mM
Done

Line 253: I'm wondering if you have any evidence that the PhoX you are quantifying from Prochlorococcus actually is a bona fide alkaline phosphatase (it is quite a different sequence (25% identity to the Syn WH8102 PhoX which has been biochemically characterised).

The *Prochlorococcus* PhoX is the below sequence and specifically the highlighted peptide was quantified using an isotopically labelled standard. This sequence has a 40% sequence identity and expectation value of 9e-148 to the extremely well characterized E.coli PhoX, which is how we initially annotated it. We agree with the reviewer that the caveats of homology-based annotation should be laid out very clearly. We have included this caveat in the introduction of the manuscript to prime the reader to consider this (see line 128).

Lines 195-246: There is a detailed proteomic protocol here but there is little or no information on how protein abundance is normalised. The only place I could find something was in the legend to Fig 4. With this in mind please can you write a more detailed normalisation protocol in the methods section (and explain in detail what you mean by nSC). You state this represents the spectral counts normalized to the maximum value of each protein across 6 stations, but it is unclear to me what this actually means. Clearly for the data presented in Table 2 and used throughout the manuscript, you want fold change in protein abundance not just to reflect cell number of Syns/Pros or total Syn/Pro protein abundance (otherwise changes in abundance of proteins merely reflect changes in the abundance of organisms rather than reflecting potential expression levels as a function of the environment). With this in mind for normalisation, please can you also inform readers how you account for changes in gene presence/absence in Pro and Syn strains (clades/ecotypes) as you move along the transect. Some alkaline phosphatases are sporadically present in these organisms and if there is no gene present clearly you won't encode protein to detect. In other words does the taxonomic composition (clade/ecotype of Syn/Pro) across the transect significantly change? You could assess this from the metagenomes you have from the cruise. (I'm guessing that Po populations are all HLII and Syn mostly clade III but some evidence for this across the transect would certainly be useful, otherwise your proteomic data could be very hard to interpret.

Thank you for this comment. There are two types of proteomics data being reported in this manuscript (non-targeted discovery metaproteomics and targeted quantitative analysis of select protein targets). This comment seems to refer to the former. For the discovery metaproteomics analysis, protein concentration data is reported as relative abundance (normalized spectral counts). The normalization is performed by summing the total number of spectra in each sample, calculating the average number of spectra across all the samples, and then multiplying each spectrum count by the average count over the sample's total spectral count. This is done to control for small differences in the amount of sample injected into the mass spectrometer. The normalized spectral count data for a given protein can therefore be interpreted as a 'fraction' of the total protein in the sample. The data as-such will indeed reflect changes in the abundance of organisms in addition to changes in protein abundance within that organism. Indeed, we can see changes in the taxonomic composition

of the microbial community reflected in the metaproteomics data (Figure 4D-F). Based on the data, the alkaline phosphatase/biomarker proteins are anti-correlated with organism abundance as reflected in the metaproteomics data (Figure 4D-F) as well as in cell count data (Figure 3) which provides an independent measure of biomass. This indicates that the abundance patterns of these proteins are not due to changes in organism abundance and rather due to biological regulation (see also around line 375).

In terms of clades/ecotypes, we do recognize and mention in the manuscript that shifts in taxonomic composition/ecotypes could alter expectations about alkaline phosphatase abundance (e.g. lines 420). Pro HLII is indeed very much the dominant ecotype across the transect (Figure S3B) being more than 10x more abundant based on the global metaproteomics data than HLI throughout the transect, even as HLI increased in abundance in the West. The fact that we were able to identify Pro and Syn PhoA and PhoX proteins across the transect strongly indicates that the genetic potential for those enzymes were present across the basin. In addition, 'patchiness' in alkaline phosphatase genetic capability seems to require quite significant differences in phosphorus biogeography (e.g. Martiny et al., 2006, though we notice the patterns in this study have been debated), whereas the entire basin that we sampled was phosphate limited, allowing us to assume eco-evolutionary controls that maintain alkP genes within these populations.

We have added the following statement ot the end of the Methods section 2.3:
Global metaproteome protein abundances are reported in normalized spectral counts.  The normalization is performed by summing the total number of spectra in each sample, calculating the average number of spectra across all the samples, and then multiplying each spectrum count by the average count over the sample's total spectral count. This is done to control for small differences in the amount of sample injected into the mass spectrometer.

Lines 257-276: Please indicate here the length of time the nutrient bioassays were carried out for. Also, please indicate when samples for proteomics were taken (I assume 48 hours). We have inserted text to note that the incubations were carried out for 48 hours.

Line 336 (Table 2): Please can you indicate a statistical significance to these fold changes e.g. is a 1.3 fold change statistically significant?
We have performed Wilcoxon test (non-normal data) to compare the west and east properties and inserted symbols (* or **) to indicate significance. Note that we cannot perform statistical analysis on the protein data because there is only one value in the east or west – proteomics were not replicated from in-situ measurements.

Line 340-1: should read between west and east (i.e. delete west after east).
Done

Line 349-350: I don't follow this statement. Fig S2 shows data as far as I can tell only for the Pro HLII ecotype and not any other clades. Also, what is the difference between the hatched and filled box for the HLII ecotype? Additionally, what is HIII? MIT9314 is a strain not a clade.

Thank you for identifying the problem with this figure. First, HIII was a typo (meant to be HII) and the hatched/filled boxes in Figure S2B show two different PhoA proteins. We have corrected these mistakes in the figure caption and the figure legend and elaborate more below.

Figure S2 shows data breaking down peptide amino acid specificity for different proteins. We use this to understand which organisms are contributing to the protein pools. Depending on which peptides are identified, we can assign taxonomic 'ownership.' Sometimes the peptide

sequences are very specific (e.g. to a specific strain like HLII strain MIT9314). Other times, the peptides are not specific to a given strain but are shared across the ecotype (the HLII peptides) or organism ( 'All Prochlorococcus') (Saunders et al., 2021). The different bar hashings represented different examples of PhoA or UrtA proteins that were identified, and we regret that this was not clearer in the original version and have corrected it now.

*For Pro-PstS and Pro-PhoA, the zonal trends in protein abundance were consistent no matter what level of taxonomic composition is considered (e.g. specific strain level, ecotype level, or organism level) for PstS (Fig. S2a), PhoA (Fig. S2b) and PhoX (Fig. S2c). The repeatability of the pattern regardless of which taxonomic level is considered reflects true biological regulation within the entire* Prochlorococcus *community, rather being contingent on variation in the abundance of one clade/strain across the transect.*
Figure caption for S2 is now correct too.

[Figure]

Figure S2. Figure S2. Zonal variation in the spectral counts of (a) PstS, (b) PhoA, (c) PhoX and (d) UrtA at different levels taxonomic specificity in *Prochlorococcus* indicating the coherence in patterns in biomarker proteins across clades. HLII 9314 peptides are those specific to the strain MIT9314 based on amino acid sequence and comparison to a comprehensive collection of isolate genomes and MAG genomes. HLII peptides are those that are specific to the HLII ecotype but are shared among many HLII strains. All Prochlorococcus peptides are those that are specific to Prochlorococcus but are shared across ecotypes.

Line 351: Please clarify what you mean by zonal trend. (I agree generally the spectral counts follow the increase in Pro cell abundance but they don't capture the oscillations in Pro cell abundance at the east end of the transect).

Pro cell abundance was measured every 2 to 4 hours, allowing oscillations in surface ocean cell abundance to be captured (Figure 3b). Samples for proteomics were collected less frequently and only at stations, so it is not possible to capture the oscillations in Pro cell abundance in the proteomics data.

We have changed the text to the following to improve clarity as follows;

*Note there was an eastward increase in total Prochlorococcus protein (reported as total spectral counts, Fig. 4d) alongside Prochlorococcus cell abundance (Fig. 3b) suggesting that untargeted metaproteomics analysis can capture trends in microbial community structure.*

Lines 353-4: This statement about Pro-PstS and Pro-PhoA is also only true if all Pro cells across the transect possess the gene in question.

We have added the following to caveat this statement:

Assuming all *Prochlorococcus* cells across the transect possess both genes then the higher *Pro*-PstS and *Pro*-PhoA in the west where there was lower *Prochlorococcus* reflects a physiological response to the nutrient environment rather than reflecting changes in biomass.

Lines 397_399: This explanation is probably fine but most Syns and some Pros also possess PtrA (as well as the PhoBR system) (Ostrowski et al., 2010). Whilst PtrA appears to be controlled by PhoBR it's also possible PtrA could respond independently (& hence perhaps to DOP directly).

We have added this point to this section as follows:

*This DOP effect was likely the result of DOP conversion to phosphate and negative regulation of the Pho operon that controls both PstS and PhoA rather than DOP directly interacting with the regulatory system, or alternatively that there is another regulatory system that is directly regulating based on DOP availability (PtrA being one alternative phosphate-sensitive regulator identified in some Synechococcus and Prochlorococcus strains and that may be responsive to organic P (Ostrowski et al., 2010).*

Line 400-1: It's probably worth to more explicitly explain what you mean here - so is this a hint that you think Pro PhoX may require zinc (not Fe)?

The factors (phosphate, metals etc) that regulate PhoX in *Prochlorococcus* are currently unknown. We observed an increase in PhoX after the addition of zinc, but don't know why. We do not think we should speculate on the potential for zinc to control PhoX.

Line 402: total Pro abundance increased across the transect from west to east (Fig 3b)..(it didn't decrease).

We have removed reference to the in-situ measurements as follows:

*It is unclear if the decline in protein biomarkers, Pro-PstS and Pro-PhoA in experiments was a physiological response to elevated DOP, or due to a decline in Prochlorococcus biomass.*

Line 403: The Pro and Syn cell number data in Table S5 is a single value (we would need to see the data for cell abundance across the time course of the bioassay). Please can this info be added to the Table (i.e. flow cytometry data for the initial time point and the 48 hour time point - which is when I assume the proteomics samples are taken).

We have now updated Table S5 with the cell abundance for Pro and Syn at Tzero, also for chlorophyll and APA. Note we do not have data for proteins from Tzero and instead compare to the control.

Line 406: you mention about targeting specific clades here – but HLII refers to an ecotype (are you saying you can target specific sub-clades within the HLII ecotype?

This was an error. I have changed clade to ecotype

Line 422: please delete psip1 here – it is not a regulator but shown to be a high affinity alkaline phosphatase (Torcello-Requena et al., 2024)

We have deleted reference to psip1 here.

Lines 423-426: It might be worth mentioning that Pro MED4 possesses both PhoBR and PtrA (though not sure if these genes are present in all HLI Pros).

We have inserted the following sentence:

This contrasts with HL1 (MED4), which possesses both regulatory genes involved in phosphorus metabolism, phoBR and ptrA (Martiny et al., 2006)

Line 431: You mention that HLI abundance typically increases with depth but also HLI abundance also generally increases in slightly higher latitude cooler waters (see AMT transect data in Johnson et al., 2006; Zwirglmaier et al., 2007, 2008 Env Micro).

We have added a statement to the end of the sentence to reflect the change with latitude too.

Lines 438-440: It is surprising that Syn-PstS was not detected in the metaproteomes since PstS is generally highly expressed in these organisms in P-deplete conditions. It's thus also odd that in contrast PstS can be detected in the bioassays – especially since from Table S3 Syn PstS is only detecting PstS from clade X which is generally not an abundant lineage in this water type (especially compared to clade III). Any thoughts on this contradiction would be useful to add.

This section is describing results of the global metaproteomics data. The method for this analysis are fundamentally different from the quantitative proteomics analyses performed for the bioassays and reported in Table S3. The global metaproteomics data was collected in data-dependent acquisition (DDA) mode in which the mass spectrometer identifies peptide ions, selects abundant ions from an MS1 scan, and then fractionates and re-analyses that selected peptide in an MS2 scan. The MS2 scan data is what is used for peptide/protein identification by bioinformatics software. This method can produce fairly comprehensive metaproteomics analyses (in our case, identifying more than 65,000 proteins) but by nature does not identify all of the proteins in the sample, being more focused on abundant peptides. Causes for not seeing an expected protein/peptide can include low protein/peptide abundance, protein localization (e.g. membrane-associated proteins), differences in peptide ionization efficiency, and co-elution with other abundant peptides. On the other hand, in the targeted analysis, the mass spectrometer is instructed to select peptides for MS2 analysis based on their inclusion in a target list; this ensures that the peptide will always be selected for analysis, even at very low abundance, and also minimizes any co-elution problems. We targeted the clade X peptide because we identified it in a prior, non-targeted analysis of experiment samples (see the companion manuscript, Held et al., 2025). While it is possible

and even likely that there are other *Synechoccocus* PstS proteins in the samples, we unfortunately cannot say anything about them given that they were not identified.

Line 451: Regarding the fold change in abundance of PstS in the bioassays do you detect the PstS peptide initially (i.e. before nutrient addition) or are you interpolating a number here to give these fold change values? Please add some info here to explain this.

Fold-changes reported for the PstS peptide (and others) are compared to the control (no nutrients added), not the initial peptide concentration. We compare to the control to take account for the effect of containment on peptides.

We have added the following text to clarify:

*relative to the control after 48-h bioassays*

Line 455: You are stating here that Syn PhoA possesses Zn or Co but we don't know this – I would change considering to assuming.

We have changed to the following:

assuming PhoA contains Zn or Co, and not Fe as metal co-factors

@CM Line 468-469: The Waterbury et al., 1986 ref doesn't show any data for alkaline phosphatase so please change this citation to one that supports this statement.

We have removed reference to Waterbury et al 1986 and included Cox and Saito 2015.

Line 469: in the presence
We have removed this second part of the sentence – it was in excess.
*suggesting upregulation of genes encoding for AP in the presence of external organic P (Moore et al., 2005).*

Lines 501-506: please delete these lines from the text since the info here relates to a different submitted paper and no cobalt additions were made in the nutrient bioassays reported here. (As an aside nutrient additions only indirectly report metal requirements of enzymes. To prove this the protein(s) would need purifying and the precise metals required for activity determined).

We will respectfully keep these lines because they relate to the companion paper of this manuscript, which was published as pre-print in the same journal shortly after the initial publication of this manuscript's preprint (https://egusphere.copernicus.org/preprints/2025/egusphere-2024-3996/). The statement has already been caveated (see below) to reflect uncertainty, however field observations can be useful first-steps in biochemical characterisation of novel enzymes.

*Caveat: implying that Co may effectively substitute Zn at the active site of PhoA within marine cyanobacteria,*

Line 522: as well as the different depths sampled you are also not capturing the whole Syn population in the bioassay experiment (since the peptide targets clade X) and this needs mentioning here.

We have added the following text to highlight the different clades targeted using quantitative peptide analysis.

These differences are likely due to the different depth horizons sampled (40m for experiments, 15 meters for metaproteome analysis) as well as the different *Synechococcus* populations captured using quantitative peptide analysis (see Table S3, clade III and X) compared to metaproteomes.

Line 534: I would be tempted to be slightly less robust that PhoA is regulated by zinc - given that the Zur regulon in WH8102 (the same strain for the Cox and Saito manuscript) does not include phoA (Mikhaylina et al., Nature Chem Biology 2022) - so there appears not to be direct regulation by a zinc responsive sensor. That said, BmtA is regulated by Zur in WH8102 and it was proposed by Mikhaylina et al., that this may provide zinc for PhoA. The authors also need to cite Ostrowski et al., 2010 since PhoA (SYNW2390) is regulated by PtrA.

While the biochemical regulatory mechanisms are not fully clear, the experimental evidence of Cox & Saito do show that PhoA is responsive to zinc and indeed that zinc seems to modulate the regulatory response to low phosphate (when no zinc is added to the media, the protein does not increase in abundance even when phosphate is low). Cox & Saito proposed a potentially indirect regulatory mechanism, which is still being chased and may indeed be the Zur/BmtA system. We have cited Ostrowski et al 2010.

Lines 537-541: As far as I can see there is no cobalt nutrient addition data included in this manuscript but it seems the submitted manuscript contains this info. Can the latter manuscript alone not discuss the cobalt data otherwise it's a bit confusing where to access this info?

We would like to keep the cobalt story in this manuscript as it compliments the results and discussion related to the role of zinc in alkaline phosphatase activity. We have reduced and simplified the text in this section too.

Line 549: which transporter are you alluding to here? – Please can you give an example locus ID or perhaps better a cyanorak cluster number here.

The identities of these putative Zn transporters are reported in detail in Table S2 (including the amino acid sequences). This is probably more useful than the protein accession numbers since we used protein sequences from a variety of sources with different naming conventions.

Line 567: I am very curious why the authors speculate that PhoX may be regulated by organic complex forms of Co or Zn. I think this may be dangerous to say given that Kathuria and Martiny Env Micro (2011) showed neither Co or Zn stimulated PhoX activity from Synechococcus sp. WH8102 (though it is a shame that Fe was not assessed in this latter study).

This is a fair criticism as the statement is highly speculative. We have dropped this statement.

Lines 569-570: Please can you explain a bit more what you mean when you say deconvoluting P strategies is complicated by other environmental and physiological factors that can influence their growth.

This sentence is written to lead into the next paragraph which considers other factors (e.g. atmospheric deposition, nitrogen fixation) that may influence the biogeography and nutrient uptake strategies of pico-cyanobacteria. The sentences have been edited for clarity as follows:

*Finally, deconvoluting the P acquisition strategies alongside the biogeography of picocyanobacteria and their ecotypes in situ is further complicated by other factors such as bioavailability of resources (including DOP), competition for resources with other phytoplankton groups and grazing, all of which influence growth. Additional mechanistic laboratory studies are needed to isolate and accelerate interpretation of picocyanobacterial alkaline phosphatases.*

Line 593: regarding Syn abundance across the transect do you have any evidence that there might be increased grazing pressure (or viral infection/lysis) in the east?

There are a handful of publications on grazing pressure and viral lysis from north-south transects in the Atlantic leveraging off the AMT transect. We could not find relevant studies on viral cell lysis in this region, or linked specifically to an environmental parameter. One study analysed 40 years of zooplankton biomass data from the Atlantic and found higher biomass in the east compared to the west. However, we have not added a statement about grazing and instead removed the text related to the biogeography of *Pro* and *Syn*. Reviewers commented on the length of the paper and the inclusion of speculative statements throughout. Adding a statement about grazing would have added speculation.

Lines 602 and 649: picocyanobacteria

Picocyanobacterial changed to picocyanobacteria

Line 689-691: PhoX in Syn WH8102 is SYNW1799 (see Kathuria and Martiny 2011) so SYNW0120 is not correct. Indeed, for SYNW1799 this is located next to potential iron transporters (which would be consistent with PhoX requiring iron as it does in other bacteria)! Please correct this.

This is an important mistake – SYN0120 is another putative alkaline phosphatase in Synechococcus but not the main PhoX protein that we focus on in this manuscript. Revise this sentence to discuss the PhoX that we meant to focus on.

*For Synechococcus (WH8102), the position of phoX (SYN1799) is like Prochlorococcus, is located directly next to the futAB iron ABC transport system, consistent with the iron requirement of this enzyme.*

Line 691: So in the lines above you mention single strains (MED4 for Prochlorococcus) or WH8102 for Syn. In this line you then state these genes are separated in the chromosome in the whole genus - did you check this is the case for all published Pro and Syn genomes? Otherwise you would need to qualify this statement.

Fair point; we can caveat the statement:

*The separation of phoA and phoX within the genome in both*

*Prochlorococcus* and *Synechococcus* (at least in the representative strains described above) implies their regulation may be distinct in the different organisms.

Line 738: mechanistic

Done

Mahaffey et al. conducted a proteomics analysis on the samples from the zonal transect of the North Atlantic subtropical gyre to investigate the distribution and nutrient acquisition strategies of *Prochlorococcus* and *Synechococcus*. The natural nutrient gradients across the zonal transect make this analysis interesting, as it may drive observable trends in the cell abundance of cyanobacteria and their nutrient-related protein expression (PstS, PhoA, PhoX, P-II, UrtA, and AmtB). The study tried to analyse all nutrient-related proteins, including phosphorus, nitrogen, and trace metals (e.g., Fe, Zn, Co), with a focus on phosphorus. The main finding may be that there is a transition from phosphorus stress in the west and nitrogen stress in the east of *Prochlorococcus* and *Synechococcus*. Overall, the study is interesting. However, there are several issues that need to be addressed prior to publication.

1. The manuscript is too lengthy to get the main idea. The author presented and discussed the protein changes associated with the nutrient strategies of *Prochlorococcus* and *Synechococcus*, including phosphorus, nitrogen, and trace metals (e.g., Fe, Zn, Co). Each section is discussed in detail, which makes the article appear unfocused. Also, it is vague what the focus of each section is. I strongly recommend restructuring the manuscript to highlight the most significant findings (specifically, the P stress and uptake strategies), and the other parts could be presented as auxiliary or as support for the main findings. In addition, the manuscript is so wordy that it is easy for readers to lose interest and patience. I suggest rewriting the Results and Discussion part to make it more concise.

2. The entire article is riddled with speculation, and the viewpoints are rarely supported by solid evidence. For example, starting from line 635, the authors discuss the impact of aerosol dust deposition on the distribution of *Prochlorococcus* and *Synechococcus,* building on many previous findings. However, the current study did not provide any evidence indicating the presence of aerosol dust during sampling. The discussion on copper toxicity is also suppositional. In fact, the whole discussion on the distribution of the cyanobacterial cell abundance is hardly convincing. The abundance of cyanobacterial cells is a result of various biological and ecological processes, including growth, competition, grazing, and mortality. Although nutrient uptake strategies and ambient nutrients significantly influence growth, the relationship between growth and cell abundance is not directly proportional. And the correlation results are not direct evidence. Therefore, I don't like this part and suggest removing it from the discussion.

3. The conclusion is too lengthy. The first paragraph is entirely unnecessary as it is repeated in the introduction and discussion sections. The conclusion should highlight the most significant findings of the study and their potential implications. Please rewrite it.

4. The abstract is also wordy. I think it exceeds the required word count of most journals.

5. Many statements in the manuscript cannot stand up to scrutiny. For instance: in line 658, "the view that the growth of picocyanobacteria are insensitive to nutrient availability". This statement is strange and misleading. The ability of cyanobacteria to thrive under ultralow nutrient conditions does not indicate nutrient insensitivity; rather, it highlights the challenge of

quantifying nutrients at such trace concentrations. in line 603, "Prochlorococcus are major players in the microbial loop and thus the availability of recycled nutrients such as ammonium and urea, driven by basin-scale processes, may influence their biogeography." The causality of this sentence is strange. The availability of recycled nutrients impacts the biogeography of *Prochlorococcus*should be because they are nutrient sources for *Prochlorococcus*?

We have taken onboard the comments regarding the length of the manuscript, lack of focus, caveats and speculation and repetition. We have edited the manuscript significantly and removed about 25% of the text.

6. It is better to reorder the figures. Fig. 1g and 1h should be put to Fig. 3 as it is biology-related. And Fig. 1h was mentioned after the Fig. 2 in the main text.

We have switched the order of the figures as suggested.

---

## Author Response (AR2)

We would like to thank the reviewer for their time in reading the revised version of the paper and for the positive feedback on the new version. We have responded to the minor comments below and have made the relevant edits on the manuscript.

The methods section does not describe the enzyme kinetic bioassays used to determine the Vmax and Km values presented in Figure 8. As these details are critical for interpreting the results, I recommend describing the assay methodology in the main text and providing the full experimental details in the supplementary materials.

We have added the following text to lines 170 to 175:

Enzyme kinetic parameters were determined at each station by incubating unfiltered surface seawater with various concentrations of the synthetic fluorogenic substrate 4-methylumbeliferyll-phosphate (MUFP, Sigma Aldrich) and measuring the change in fluorescence for 8 hours (as described by Davis et al., 2019). The maximum hydrolysis rates ($V_{max}$) and the half saturation constant ($K_m$) were determined using the Hanes-Woolf plot graphical linearization of the Michaelis-Menten equation following Duhamel et al. (2011).

2. The influence of nitrogen acquisition on the biogeography of Prochlorococcus and Synechococcus has not been comprehensively discussed. While the study describes the spatial patterns of nitrogen stress in Prochlorococcus and Synechococcus, the direct influence of this stress on their biogeographical distribution and abundance is not sufficiently explored. A more thorough discussion explicitly linking the observed N stress to the control of population dynamics is needed.

We have revised Section 3.5 (from line 611) to read as follows:

The spatial patterns in nitrogen stress biomarkers gleaned from non-targeted metaproteomics provided insight into how fixed nitrogen availability contributed to shaping the biogeography of *Prochlorococcus* and *Synechococcus* across the subtropical Atlantic. Surface ocean gradients in fixed nitrogen are established via a combination of upwelling in the eastern Atlantic (Menna et al, 2015), nitrogen fixation (Fig. 3g, Cerdan-Garcia et al., 2022) and dust deposition (Powell et al., 2015) delivering nitrate, ammonium and urea to the surface subtropical Atlantic Ocean alongside microbial demand consuming fixed nitrogen. In summer 2017, concentrations of nitrate (< 40 nM) and ammonium (< 20 nM) were relatively low across the transect (Fig. 1e and f). The lowest spectral counts of the N-stress proteins, AmtB and UrtA in *Prochlorococcus* (Fig. 4c) coincided with a region of elevated nitrogen fixation rates (Fig 3g), reflecting alleviation of N stress in *Prochlorococcus*. The combination of efficient uptake of nitrogen derived from nitrogen fixers (Caffin et al., 2018), alongside the small cell size of *Prochlorococcus* provides a strong competitive advantage under oligotrophic conditions, as observed in the North Pacific gyre (Saito et al., 2014, 2015).

Otherwise, *Prochlorococcus* was subjected to increasing N stress towards the eastern Atlantic, evidenced by eastward increases in protein biomarkers in *Prochlorococcus*, specifically P-II, ammonium transporter AmtB and urea transporter UrtA (30-70%, Fig. 4b and Table 2).

In contrast to *Prochlorococcus*, consistently elevated UrtA in *Synechococcus* suggests chronic N stress throughout the transect, with UrtA in *Synechococcus* being more than 5 times higher than for *Prochlorococcus* (Fig. 4b). This is consistent with the physiological disadvantage of *Synechococcus*, with its larger cell size and less efficient surface-area to volume ratio for nutrient acquisition (Chisholm 1992). P-II and AmtB (or NtcA) was not detected in the metaproteome of *Synechococcus*, perhaps because *Synechococcus* was 5 to 10 times less abundant in the metaproteomes compared to *Prochlorococcus*. The dominance of proteins for ammonium and urea acquisition of *Synechococcus* and *Prochlorococcus* are consistent with the premise that while marine *Synechococcus* and some *Prochlorococcus* strains have the genetic makeup to assimilate nitrate (Berube et al., 2015; Domínguez-Martín et al., 2022; Martiny et al., 2009), it accounts for < 5% of their total N demand, and instead ammonium and urea are the dominant N sources (Berthelot et al., 2019; Casey et al., 2016; Painter et al., 2008). The patterns observed align with established biogeographical trends in which *Prochlorococcus* dominates in the nutrient-deplete surface ocean due to its competitive advantage as a small cell, whereas Synechococcus persists in regions where fixed N is available. The proteomic data indicate that nitrogen acquisition traits are one of the key determinants of population dynamics, driving spatial partitioning between *Prochlorococcus* and *Synechococcus* and ultimately influencing primary productivity and nutrient cycling across the subtropical Atlantic.

This manuscript is part 1 in a two-part publication from a project on the prevalence of zinc and iron limitation in alkaline phosphatase in the subtropical ocean. As such, we have been advised to add a pre-title to both manuscripts and therefore the title has been edited to the following:

**Proteomic and biogeochemical perspectives on cyanobacteria nutrient acquisition:** **Part 1: Zonal gradients in phosphorus and nitrogen acquisition and stress revealed by metaproteomes of** ***Prochlorococcus*** **and** ***Synechococcus***